# MIND THE GAP: GLIMPSE-BASED ACTIVE PERCEPTION IMPROVES GENERALIZATION AND SAMPLE EFFICIENCY OF VISUAL REASONING

**Oleh Kolner**[1,2]**, Thomas Ortner**[1]**, Stanisław Woźniak**[1] **& Angeliki Pantazi**[1]
[1]IBM Research Europe – Zurich, Switzerland
[2]Institute of Machine Learning and Neural Computation, Graz University of Technology, Austria
`olk@zurich.ibm.com`

## ABSTRACT

Human capabilities in understanding visual relations are far superior to those of AI systems, especially for previously unseen objects. For example, while AI systems struggle to determine whether two such objects are visually the same or different, humans can do so with ease. Active vision theories postulate that the learning of visual relations is grounded in actions that we take to fixate objects and their parts by moving our eyes. In particular, the low-dimensional spatial information about the corresponding eye movements is hypothesized to facilitate the representation of relations between different image parts. Inspired by these theories, we develop a system equipped with a novel Glimpse-based Active Perception (GAP) that sequentially glimpses at the most salient regions of the input image and processes them at high resolution. Importantly, our system leverages the locations stemming from the glimpsing actions, along with the visual content around them, to represent relations between different parts of the image. The results suggest that the GAP is essential for extracting visual relations that go beyond the immediate visual content. Our approach reaches state-of-the-art performance on several visual reasoning tasks being more sample-efficient, and generalizing better to out-of-distribution visual inputs than prior models.

## 1 INTRODUCTION

Humans use vision both to detect objects and to analyze various relations between them. For example, to successfully grasp an object from a table, we should first see whether some other object is not lying on top of it. Such relations can be spatial, like in the aforementioned example, or more abstract, e.g. if two objects are the same or different. While humans can easily recognize many kinds of relations, even between previously unseen objects (Fleuret et al., 2011; Kotovsky & Gentner, 1996), a growing body of literature suggests that artificial neural networks (ANNs) struggle with this (Kim et al., 2018; Ricci et al., 2021; Puebla & Bowers, 2022; 2024). Typically, ANNs process visual information holistically with each part of an image being processed in parallel with the others. In contrast, humans use active vision by selectively moving their eyes to glimpse at salient and/or task-relevant parts of the image and to process those parts in high resolution (Yarbus, 1967; Hayhoe & Ballard, 2005; Land, 2009). Moreover, by moving the eyes, the brain is able to form representations of the corresponding glimpse locations. This process can also be seen as a factorization of the visual scene into its visual ("what") and spatial ("where") contents (Goodale & Milner, 1992; Behrens et al., 2018). It is hypothesized that the low-dimensional representation and the relational geometry between those locations form a scaffold for representing the image structure, i.e. relations between different image parts (Summerfield et al., 2020).

Currently, the best-performing ANNs, proposed for visual reasoning, use completely different mechanisms compared to humans (Webb et al., 2024a; Mondal et al., 2024). Specifically, they combine an object-centric representation using slot attention (Locatello et al., 2020) and a special inductive bias called relational bottleneck (Webb et al., 2024b). The former aims at decomposing an image into a set of representations corresponding to its objects and the latter encourages the model to focus on the relations between objects' representations instead of the representations themselves. While showing

compelling performance on several visual reasoning tasks, these ANNs struggle to generalize the learned rules to out-of-distribution (OOD) objects (Puebla & Bowers, 2024). A likely reason is that the methods for object-centric representation cannot reliably capture the relations between different parts of objects. Instead, they merely perform perceptual segregation through clustering (Mehrani & Tsotsos, 2023). Inspired by active vision, an alternative model, proposed by Vaishnav & Serre (2023), processes images iteratively using a soft attention mechanism controlled by a memory module. However, in every iteration, this model still processes the image in its entirety only masking specific regions. Moreover, it lacks a low-dimensional spatial representation of task-relevant regions.

Drawing more inspiration from humans's active vision, and the aforementioned theories behind it, we introduce a model equipped with a novel Glimpse-based Active Perception (GAP) that sequentially glimpses at the most salient parts of the image. The GAP produces a dual sequence of glimpse locations and glimpse contents around them, promoting spatial "where" and visual "what" pathways. Importantly, the glimpse locations in the "where" pathway convey purely spatial information that is complementary to the visual contents of the "what" pathway. In particular, the geometrical relations between those locations provide a basis for extracting structural aspects of the image that can also be present in an image with different visual contents. The GAP can be paired with various downstream architectures that can process the dual sequence to solve the task at hand. In this work, we use a Transformer (Vaswani et al., 2017) and its recently introduced extension, called Abstractor (Altabaa et al., 2024). We evaluate our model on four visual reasoning datasets testing OOD generalization capabilities and sample efficiency. Achieving state-of-the-art performance, our model provides computational support for the active vision theories.

Our contributions can be summarized as follows:

- We present a brain-inspired active vision system for visual reasoning that extracts structural aspects of images by sequentially glimpsing at their salient parts.
- We show that our system outperforms state-of-the-art models on several benchmarks in terms of sample efficiency and OOD generalization.
- Our model is amenable to the integration of more advanced components for both saliency detection and downstream architecture which allows further scaling and extension.
- Our results support cognitive science theories postulating that the brain factorizes images into their visual and spatial contents with the latter being crucial for OOD generalization.

## 2 APPROACH

We introduce a model that learns to extract relations between different parts of an image by factorizing it into its visual ("what") and spatial ("where") contents. This is achieved by sequentially glimpsing at the most salient image parts that comprise the visual content whereas the locations of those parts provide the spatial content. Our approach is illustrated in Figure 1. Given an input image, our model employs a novel concept of error neurons to extract a saliency map where stronger activities of the error neurons correspond to more salient parts, such as corners of the shapes presented in the image. The saliency map is then used to guide the glimpsing process by producing locations to the highest saliency, referred to as glimpse locations, in a winner-takes-all (WTA) fashion. To prevent re-visiting previous glimpse locations, an inhibition-of-return (IoR) mechanism updates the saliency map by masking out regions around the previously visited locations. These two steps are inspired by (Itti et al., 1998). Each glimpse location is passed to a glimpse sensor that extracts the corresponding visual content around that location, referred to as glimpse content. The described process repeats for a pre-defined number of iterations, producing a sequence of glimpse contents and a sequence of glimpse locations. These two sequences represent the factorization of the image into its complementary "what" and "where" contents. Both sequences are processed by the downstream architecture that makes the final decision for the given the task.

### 2.1 GLIMPSE-BASED ACTIVE PERCEPTION

Glimpse-based Active Perception (GAP) is a process of acquiring visual and spatial information by steering a glimpse sensor to different parts of the visual scene. The steering is facilitated by a saliency map that highlights the most salient image parts. While there are many ways to compute such a saliency map, we use a simple but efficient concept of error neurons (Figure 2(a)) described

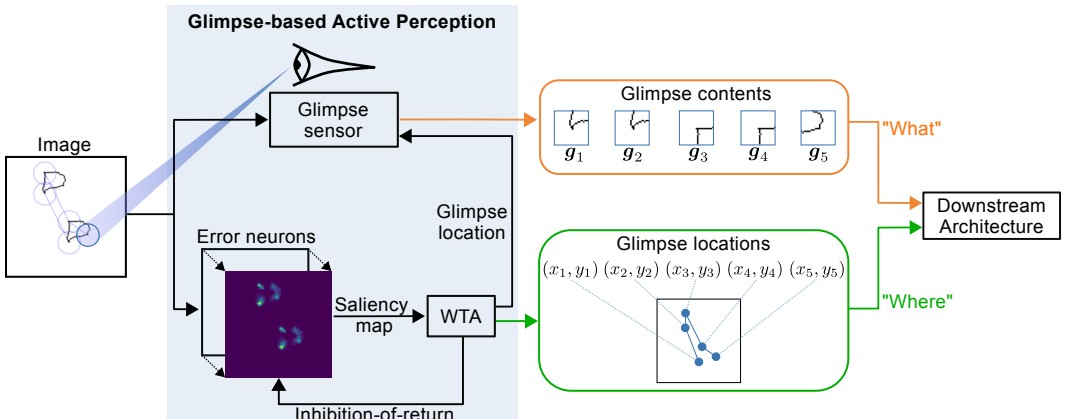

Figure 1: **Architecture overview.** Based on the input image, the layer of error neurons produces the saliency map where salient image parts are highlighted (brighter regions). Through WTA combined with inhibition-of-return, a sequence of glimpse locations is produced that covers the most salient regions. For each such location, the glimpse sensor extracts the glimpse content from the region around it. The sequences of glimpse locations and glimpse contents are processed by the downstream architecture to make the final decision about the presented task.

in detail in Section 2.1.1. Based on the saliency map $\boldsymbol{S} \in \mathbb{R}^{h_S \times w_S}$, $h_S, w_S$ denote the saliency map's height and width, the model glimpses at the most salient regions for a pre-defined number of iterations $T$. Specifically, at each iteration $t$, the next glimpse location $\boldsymbol{x_t} \in \mathbb{N}^2$ is selected through WTA competition returning the 2D location of the highest saliency:

$$\boldsymbol{x_t} = \text{WTA}(\boldsymbol{S_t}) \stackrel{\text{def}}{=} \arg\max_{ij}(S_{t,ij}), \tag{1}$$

where $\boldsymbol{S_t}$ is the saliency map at iteration $t$. To prevent the same location from being selected at subsequent iterations, the IoR mechanism applies a mask $\boldsymbol{M}(\boldsymbol{x_t}) \in \mathbb{R}^{h_S \times w_S}$ around this location, updating the saliency map:

$$\boldsymbol{S_{t+1}} = \boldsymbol{S_t} \odot \boldsymbol{M}(\boldsymbol{x_t}). \tag{2}$$

The mask $\boldsymbol{M}(\boldsymbol{x_t})$ can be either a hard or a soft one. The hard mask consists of ones except for a round region of zeroes around the location $\boldsymbol{x_t}$ as shown in red in Figure 2(b). The size of that region is a hyper-parameter. In the soft mask, the values at each mask location $i, j$ are computed via the radial basis function with the Gaussian kernel $e^{-\epsilon||\binom{i}{j}-\boldsymbol{x_t}||_2}$ with $\epsilon$ being a hyper-parameter and $\binom{i}{j} \in \mathbb{N}^2$ a vector of location $i, j$.

Given a glimpse location $\boldsymbol{x_t} = (x_t, y_t) \in \mathbb{N}^2$, the corresponding region – glimpse content – is extracted by sampling pixels from the area around it. This sampling process is conceived as obtaining information from a glimpse sensor that provides a limited view mimicking thereby a human eye that fixates different parts of the visual surroundings. The main goal of such a glimpse sensor is to strike a balance between the size of the glimpse content and the amount of information to be captured from the image. In this work, we implement two kinds of sensors: a multi-scale sensor and a log-polar sensor. First, the multi-scale sensor, introduced in (Mnih et al., 2014), extracts several patches of the same size $h_g \times w_g$, but with different resolutions, around the glimpse location, as illustrated in Figure 2(c) (three patches are pictured). The resulting glimpse content $\boldsymbol{g_t} \in \mathbb{R}^{s_g \times h_g \times w_g \times c_I}$ is the concatenation of those patches where $s_g$ is the number of different scales supported by the multi-scale sensor and $c_I$ is the number of channels in the input image. The patch with the highest resolution corresponds to the patch cut out from the input image at the glimpse location in the original resolution. The other patches are of decreasing resolution and correspond to larger areas of the input image that are down-scaled to $h_g \times w_g$, allowing to capture coarse information of the surround of the glimpse location. Second, the log-polar glimpse sensor is inspired by the human foveal vision (Schwartz, 1977; Weiman & Chaikin, 1979; Javier Traver & Bernardino, 2010) and samples pixels according to a distribution defined by a log-polar grid, illustrated in Figure 2(d). This means that more pixels are sampled closer to the glimpse location, and fewer pixels are sampled farther from it, producing a glimpse content $\boldsymbol{g_t} \in \mathbb{R}^{h_g \times w_g \times c_I}$ whose size is equal to a single image patch.

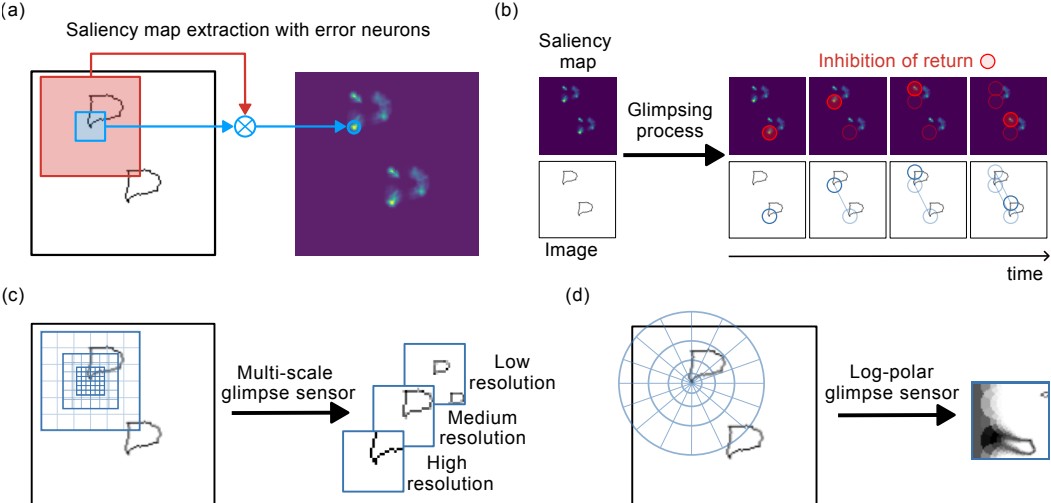

Figure 2: **Components of Glimpse-based Active Perception**. (a) Error neurons: a sample error neuron, marked by the blue circle in the saliency map on the right, computes the difference between the visual content in its central receptive field (blue square) and the visual content in its surroundings (area shaded in red). (b) Glimpsing process: at each iteration the WTA mechanism selects a location of the highest value in the saliency map (corresponding to its brightest regions) that is passed to the glimpse sensor to extract the corresponding glimpse content (blue circles). Afterward, the inhibition-of-return mechanism masks out the selected location in the saliency map (shown by red circles) so that it does not get selected again. (c) Multi-scale glimpse sensor outputs a concatenation of several patches (three are shown) of the same size each of which encompasses a region of different size and resolution around the glimpse location on the left (depicted by three grids). (d) Log-polar glimpse sensor samples pixels according to the log-polar grid shown on the left providing high resolution towards its central region and low resolution towards the periphery. After the sampling process, the polar grid is warped into a regular rectangular grid which results in the glimpse content on the right.

Using these procedures, the GAP ultimately produces a sequence of glimpse contents $\boldsymbol{g_1}, \boldsymbol{g_2}, \ldots, \boldsymbol{g_T}$ and a sequence of glimpse locations $\boldsymbol{x_1}, \boldsymbol{x_2}, \ldots, \boldsymbol{x_T}$, that are further passed to the downstream architecture described in Section 2.2.

### 2.1.1 SALIENCY MAP EXTRACTION WITH ERROR NEURONS

In the primary visual cortex, it is observed that the activity of neurons is influenced not only by stimuli from their local receptive fields but also by stimuli that come from the surroundings of those receptive fields (Vinje & Gallant, 2000; Keller et al., 2020). From a theoretical view, this influence can make the same visual stimulus be perceived as more or less salient, depending on how significantly it differs from its local surroundings (Treisman & Gelade, 1980). Inspired by this, we propose a computational block called error neurons, illustrated in Figure 2(a), to compute the saliency map $\boldsymbol{S} \in \mathbb{R}^{h_S \times w_S}$. Specifically, at each location $i, j$ of the saliency map $\boldsymbol{S}$ the corresponding error neuron computes the saliency as

$$\boldsymbol{S}_{ij} = \underset{l,k \in \mathrm{surr}(i,j)}{\mathrm{Agg}} \left[ D(\boldsymbol{P}_{ij}^{\boldsymbol{I}}, \boldsymbol{P}_{lk}^{\boldsymbol{I}}) \right], \tag{3}$$

where $\boldsymbol{P}_{ij}^{\boldsymbol{I}} \in \mathbb{R}^{h_P \cdot w_P \cdot c_I}$ is a flattened patch obtained from the input image $\boldsymbol{I} \in \mathbb{R}^{h_I \times w_I \times c_I}$ around location $i, j$. The patch size, $h_P \times w_P$, is a hyper-parameter, and $h_I, w_I, c_I$ denote the input image's height, width, and the number of channels. Unless specified otherwise, $h_I = h_S$ and $w_I = w_S$. $D(\cdot, \cdot)$ is a distance function, in this work, set to L2 distance. $\mathrm{Agg}[\cdot]$ is an aggregation operator such as sum or minimum over distances between the patch at location $i, j$ and the patches from surrounding locations $l, k \in \mathrm{surr}(i, j)$. The set of surrounding locations, $\mathrm{surr}(i, j)$, can be defined arbitrarily, comprising, for example, all available image locations. In this work, we assume the 8-connected neighborhood of location $i, j$. Intuitively, each value $\boldsymbol{S}_{ij}$ represents how much the visual content around location $i, j$ differs (or stands out) from the surrounding visual content.

## 2.2 Downstream architecture

As detailed above, the GAP provides a sequence of glimpse contents $g$ and a sequence of the corresponding 2D glimpse locations $x$ to the downstream architecture. Importantly, our approach is flexible with respect to the choice of the downstream architecture. In this work, we explore two options: Transformers (Vaswani et al., 2017) and Abstractors (Altabaa et al., 2024).

Transformers have emerged over recent years as popular state-of-the-art architectures for a variety of tasks, including the vision domain with the Vision Transformers (ViT) Dosovitskiy et al. (2021). Apart from their successes in machine learning tasks, there have also been works suggesting potential links between ViT and biology (Pandey et al., 2023). The ViT processes a sequence of image patches, augmented with positional encodings, to solve the given task. In our case, we provide the sequence of glimpse contents augmented with the respective glimpse locations as inputs.

The Abstractor has been recently introduced as an extension of the Transformer, with the key difference of the so-called Relational Cross Attention (RCA) module. It was designed to disentangle relational information from the perceptual one to enable generalization from limited data. This is achieved by modifying the self-attention so that the queries and the keys are produced based on perceptual inputs whereas the values are provided by a separate independent set of representations. The latter, in our case, is substituted by the sequence of glimpse locations whereas the perceptual inputs comprise the sequence of glimpse contents. Hence, adopting the Abstractor allows for the explicit processing of spatial information so that the visual information carries only a modulatory effect in form of attention. The formal description of the Abstractor is provided in Appendix A.

## 3 Related work

Inspired by human active vision, several models for learning visual relations have been introduced. By decomposing input images into a sequence of image regions obtained from synthetic eye movements, Woźniak et al. (2023) demonstrated that visual reasoning tasks can be solved with high accuracy using relatively compact networks. However, the synthetic eye movements were based on hard-coded locations, whereas in our case, the glimpsing process is controlled by a saliency map. The approach from (Larochelle & Hinton, 2010) and its ANN-based extension (Mnih et al., 2014) use reinforcement learning (RL) to train a glimpsing policy to determine the next glimpse location either in a continuous 2D space or in a discrete grid-based space of all possible glimpse locations. While this approach was evaluated mainly on simple image classification and object detection tasks, Ba et al. (2014); Xu et al. (2015) extended it to more complex images and image captioning tasks. However, having been evaluated on several visual reasoning tasks by Vaishnav & Serre (2023), the RL-based approaches could not achieve reasonable performance. This is likely caused by learning inefficiency in exploring the large space of all possible glimpse locations. Nevertheless, the RL-based approaches that use more efficient RL techniques such as (Pardyl et al., 2025) to learn complex glimpsing policies are relevant to our work as they can be integrated into our model to enhance its capabilities of dealing with real-world images. Our approach, in contrast, leverages the concept of saliency maps to determine the next glimpse location which significantly reduces the space of glimpse locations to the most salient ones. Gregor et al. (2015) proposed a variational auto-encoder that uses an attention mechanism for an iterative (re-)construction of complex images by attending to their different parts. Its extension proposed by Adeli et al. (2023) was designed to tackle visual reasoning tasks where it outperformed baseline CNNs. However, being trained by image reconstruction, the model does not always generalize well to OOD visual inputs. Vaishnav & Serre (2023) introduced a recurrent soft attention mechanism guided by a memory module that masks specific parts of the image where the processing should be focused. At each iteration, the model can thereby attend to multiple regions of the image simultaneously which is different from our approach where only one region can be focused at a time.

Another family of models combines object-centric representations (Greff et al., 2019; Locatello et al., 2020) with relational inductive biases (Webb et al., 2021; Kerg et al., 2022; Webb et al., 2024b). The former decomposes an image into objects present in it while the latter constrains the processing to relations between objects instead of their specific properties. Webb et al. (2024a) used pre-trained slot attention (Locatello et al., 2020) to segment objects into a sequence of slots that contain disentangled representations of objects' visual features and their positional information. Each pair of slots is then processed by a relational operator and the resulting sequence of

relational representations is processed by the Transformer (Vaswani et al., 2017). The successor model, Slot-Abstractor (Mondal et al., 2024), processes the sequence of slots with a recently introduced Abstractor module (Altabaa et al., 2024) that proved to be especially effective in reasoning about relations between objects using their disentangled visual and positional features. Similar to our approach, there are two streams of information: one based on visual features and one based on positional information. However, the latter stream consists of positional embeddings that are learned during the pre-training process which may make the model more fragile to OOD inputs. In contrast, our approach uses the raw 2D locations of salient regions. Interestingly, in the domain of visually extended large language models (LLMs), Bhattacharyya et al. (2024) showed that forcing these models via surrogate tasks to collect relevant pieces of information before solving the actual task improves the final performance. While using a different approach compared to ours to achieve that, this work points to the potential of integrating our conceptual approach into LLMs.

## 4 EXPERIMENTS

**Datasets** Our approach was evaluated on four visual reasoning datasets illustrated in Figure 3. We start with an evaluation on all 23 tasks of the SVRT dataset (Fleuret et al., 2011) to see how efficiently the model can learn spatial and similarity relations between simple shapes, see an example in Figure 3(a). Next, we test the model's OOD capabilities to generalize learned relations to previously unseen shapes. Specifically, we use an extended version of the task #1 from the SVRT dataset (Puebla & Bowers, 2022), referred to as SVRT #1-OOD, see Figure 3(b). It contains test sets with OOD shapes coming from 13 different distributions. Additionally, we use the ART dataset (Webb et al., 2021), which contains more abstract relations, see Figure 3(c). Finally, we test the model's potential to scale to more complex images by testing it on the CLEVR-ART dataset (Webb et al., 2024a), see Figure 3(d). It contains two tasks from ART but with more realistic images. Each dataset is described in detail in Appendix B. While all tasks described above consist of simple images, they are still challenging, as illustrated by the limited accuracy of the baselines described below.

**Baselines** We compare the impact of our approach with the plain downstream architectures that receive entire images as inputs. The images get split into non-overlapping patches and each patch gets augmented by its positional information as is done by ViTs (Dosovitskiy et al., 2021). We also tested those models where the patches overlapped but did not observe any significant performance differences. Therefore, we only report the standard approach with non-overlapping patches. We refer to these models simply as *Transformer* and *Abstractor*. The comparison with these baselines allows us to demonstrate the importance of processing only the relevant patches obtained from the glimpsing process rather than all patches that constitute an image. We also compare our approach with GAMR (Vaishnav & Serre, 2023), ResNet (He et al., 2015), its extension Attn-ResNet (Vaishnav et al., 2022), OCRA (Webb et al., 2024a) and Slot-Abstractor (Mondal et al., 2024). To

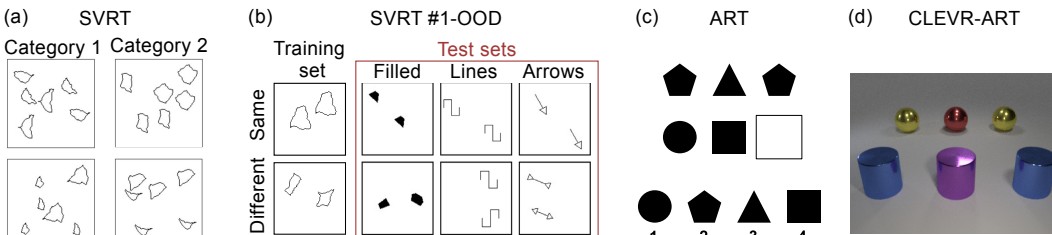

Figure 3: **Four datasets used in our experiments.** (a) SVRT dataset consists of 23 binary classification tasks where the model has to determine whether shapes are in a certain spatial and/or similarity relation to each other. Shown are images from the task of determining whether the same triplets of shapes exist in each image pair. (b) SVRT #1-OOD instantiates the first task from SVRT where it has to be determined whether two shapes are the same up to a translation. While this dataset contains training images from the original SVRT dataset, it contains 13 additional test sets (only three are shown here) with novel shapes to evaluate OOD generalization. (c) ART dataset contains four tasks where the model has to determine whether shapes in each two rows are arranged under the same abstract rule. (d) CLEVR-ART contains two tasks from ART but with more complex images.

the best of our knowledge, these baselines are the best-performing models on the tasks that we consider.

## 5 RESULTS

In all experiments, we report the performance of GAP combined with two downstream architectures, Transformer and Abstractor. The corresponding models are referred to as *GAP-Transformer* and *GAP-Abstractor* respectively. Details about hyper-parameters and training process are described in Appendix C. One important hyper-parameter is the type of glimpse sensor (multi-scale or log-polar) that we select for each of the four considered datasets. We report performance for different glimpse sensors in Appendix D. Our code is publicly available at `https://github.com/IBM/glimpse-based-active-perception`.

### 5.1 VISUAL REASONING PERFORMANCE AND SAMPLE EFFICIENCY

We evaluate the visual reasoning capabilities using the SVRT dataset that consists of 23 tasks, 11 of which require to reason about same-different relations (SD tasks) and the remaining 12 tasks require to reason about spatial relations (SR tasks). Consistent with (Vaishnav & Serre, 2023; Webb et al., 2024a; Mondal et al., 2024), Table 1 shows accuracy performance for models trained with 1000 and 500 samples. Our best-performing model, the GAP-Abstractor, outperforms the prior art models. In particular, it achieves very high accuracy on the SD tasks, considered to be more challenging than

Table 1: **Test accuracy for SVRT evaluated for common dataset sizes of 1000 and 500 samples**. The results obtained from the best trained models are averaged over SD, SR and over all (SD + SR) tasks. The standard deviation is calculated over accuracies for each individual task. Prior art results are listed in the top section whereas results obtained from our evaluations are listed in the two bottom sections. The highest and the second-highest mean accuracy in each column is marked in **bold** and underlined, respectively. The models appended by * contain a module pre-trained on the train sets from all SVRT tasks to segment the shapes.

| Model | SD tasks accuracy [%] | | SR tasks accuracy [%] | | **All tasks accuracy [%]** | |
| --- | --- | --- | --- | --- | --- | --- |
| | Dataset size 1000 | Dataset size 500 | Dataset size 1000 | Dataset size 500 | Dataset size 1000 | Dataset size 500 |
| ResNet | 56.9 ± 2.5 | 54.9 ± 2.2 | 94.9 ± 1.6 | 85.2 ± 4.3 | 76.7 ± 2.0 | 70.7 ± 3.3 |
| Attn-ResNet | 68.8 ± 4.4 | 62.3 ± 3.5 | 97.7 ± 0.7 | 94.8 ± 1.4 | 83.9 ± 2.5 | 79.3 ± 2.4 |
| GAMR | 82.1 ± 8.4 | 76.8 ± 8.7 | **98.7 ± 0.3** | **97.4 ± 0.7** | 90.8 ± 4.2 | 87.6 ± 4.6 |
| OCRA* | 90.3 ± 4.1 | 79.9 ± 4.5 | 95.0 ± 2.4 | 89.3 ± 2.5 | 92.8 ± 3.2 | 84.8 ± 3.5 |
| Slot-Abstractor* | 91.9 ± 4.0 | 82.2 ± 4.7 | 97.3 ± 1.1 | 91.8 ± 2.2 | 94.7 ± 2.5 | 87.2 ± 3.4 |
| Transformer | 51.9 ± 2.6 | 51.6 ± 1.7 | 64.8 ± 13.7 | 59.9 ± 11.0 | 56.4 ± 10.3 | 55.9 ± 8.9 |
| GAP-Transformer | 83.3 ± 13.2 | 78.4 ± 13.5 | 98.2 ± 2.2 | 97.3 ± 2.7 | 91.1 ± 11.8 | 88.3 ± 13.4 |
| Abstractor | 51.3 ± 2.5 | 51.0 ± 1.9 | 60.9 ± 14.3 | 57.0 ± 11.0 | 54.6 ± 9.7 | 54.1 ± 8.5 |
| GAP-Abstractor | **93.1 ± 13.1** | **90.5 ± 14.5** | 98.5 ± 1.9 | 96.6 ± 1.2 | **95.4 ± 8.9** | **93.9 ± 10.1** |

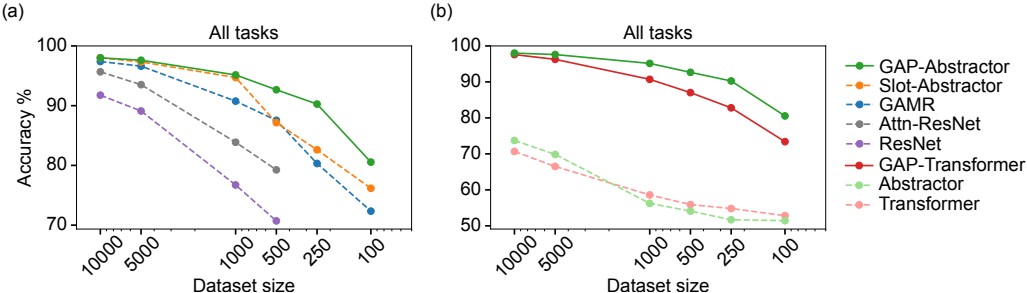

Figure 4: **Sample efficiency evaluation for SVRT beyond common dataset sizes of 1000 and 500 samples**. Average test accuracy on all 23 SVRT tasks depending on the size of the training dataset. Panel (a) compares our best model with prior art. Panel (b) compares our glimpse-based models with their ablated counterparts.

Table 2: **OOD performance for SVRT #1-OOD**. The accuracy results are averaged over all OOD test sets for 10 trained models. Prior art results are listed in the top section whereas results obtained from our evaluations are listed in the two bottom sections. The highest and the second-highest mean accuracy is marked in **bold** and underlined, respectively. The models appended by * contain a module pre-trained on the train set to segment the shapes.

| Model | Accuracy [%] (averaged over runs and OOD datasets) |
|---|---|
| ResNet | 71.2 ± 17.7 |
| GAMR | 79.8 ± 13.1 |
| OCRA* | 73.9 ± 17.2 |
| Slot-Abstractor* | 72.8 ± 16.0 |
| Transformer | 50.0 ± 0.1 |
| GAP-Transformer | 76.3 ± 18.4 |
| Abstractor | 66.2 ± 16.0 |
| GAP-Abstractor | **89.6 ± 8.7** |

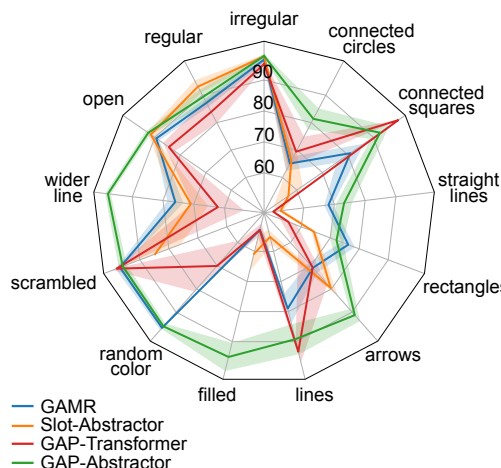

Figure 5: **Accuracy on each of the SVRT #1-OOD test sets**. The accuracy results (in %) are averaged over 10 trained models. Slot-Abstractor could not be properly evaluated on 'random color' test set that contains RGB images because its pre-trained component handles only grayscale images.

the SR tasks, for which it performs comparably with the prior art. It should be noted that although for the SR tasks, the best-performing model is GAMR, its performance on SD tasks is significantly lower compared with all other models. Detailed accuracy for each individual task is provided in Appendix, Figure D.6. Furthermore, Figure 4 shows an extensive evaluation of sample efficiency, in which we trained the models with different amounts of training data beyond those reported in Table 1 and in prior works. Compared with the results we obtained for the prior art models, our best model exhibits higher sample efficiency, especially in scenarios with fewer than 1000 training samples, see Figure 4(a). Moreover, we observe favorable trends and improvements when applying GAP to either of the downstream architectures, see Figure 4(b). More detailed results with separate performance for SD and SR tasks are provided in Appendix, Figure D.7.

## 5.2 OUT-OF-DISTRIBUTION GENERALIZATION OF SAME-DIFFERENT RELATION

While the SVRT dataset allows us to see whether a model can learn visual relations from a limited amount of training data, it does not allow us to test the model's ability to generalize those relations to OOD inputs. We evaluate the OOD generalization of a single same-different relation using the SVRT #1-OOD dataset (Appendix B.2) where a model has to determine whether two shapes are the same or different up to translation. Importantly, the models are evaluated on test sets with OOD shapes coming from 13 different distributions (see Figure B.2 for examples). Table 2 shows the performance averaged over all OOD test sets and Figure 5 shows the performance on each of the test sets. Outperforming the prior art models, our model, GAP-Abstractor, performs quite consistently on all test sets being slightly less accurate on only two sets, 'straight_lines' and 'rectangles'. Figure B.3 illustrates the reason why these two sets are more challenging.

## 5.3 OUT-OF-DISTRIBUTION GENERALIZATION OF MORE ABSTRACT RELATIONS

Going beyond evaluations of OOD generalization on a single task, we expand the evaluations to four tasks from the ART dataset, three of which contain more abstract relations (see Appendix B.3). Moreover, the evaluations on this dataset validate also the insights on sample efficiency since the models are required to be trained on very few samples. From Table 3 it can be seen that the GAP-Abstractor outperforms all other models demonstrating superior capabilities to generalize abstract relations to novel visual inputs. Crucially, this performance improvement can be observed even though no component of our model was pre-trained, as opposed to OCRA or Slot-Abstractor that have been pre-trained to segment objects.

Table 3: **Test accuracy for ART**. The accuracy results are averaged over 10 trained models. Prior art results are listed in the top section whereas results obtained from our evaluations are listed in the two bottom sections. The highest and the second-highest mean accuracy in each column is marked in **bold** and underlined, respectively. The models appended by * contain a module to segment the shapes that was pre-trained on images with shapes distinct from those in the ART dataset.

| Model | Task accuracy [%] | | | |
|---|---|---|---|---|
| | SD | RMTS | Dist3 | ID |
| GAMR | 83.5 ± 1.4 | 72.2 ± 3.0 | 68.6 ± 1.8 | 66.2 ± 4.3 |
| ResNet | 66.6 ± 1.5 | 49.9 ± 0.2 | 50.1 ± 1.3 | 54.8 ± 2.4 |
| OCRA* | 87.9 ± 1.3 | 85.3 ± 2.0 | 86.4 ± 1.3 | 92.8 ± 0.3 |
| Slot-Abstractor* | 96.4 ± 0.4 | 91.6 ± 1.6 | 95.2 ± 0.4 | 96.4 ± 0.1 |
| Transformer | 55.9 ± 2.1 | 51.3 ± 0.1 | 29.3 ± 1.2 | 33.3 ± 1.7 |
| GAP-Transformer | 76.4 ± 13.7 | 62.2 ± 7.3 | 50.7 ± 6.1 | 55.5 ± 5.8 |
| Abstractor | 70.9 ± 6.6 | 51.3 ± 0.2 | 58.9 ± 3.8 | 47.9 ± 1.4 |
| GAP-Abstractor | **97.7 ± 2.0** | **96.3 ± 1.0** | **98.4 ± 0.5** | **96.8 ± 1.8** |

## 5.4 SCALING POTENTIAL FOR MORE COMPLEX VISUAL SCENES

We test our approach on two tasks from ART – relational-match-to-sample (RMTS) and identity rules (ID) – but with more realistic images using the CLEVR-ART dataset. In the RMTS task the model has to determine whether the objects in the top row are in the same "same/different" relation as the objects in the bottom row. In the ID task the model has to determine whether the bottom row contains objects that follow the same abstract pattern (ABA, ABB or AAA) as the objects in the top row (see Figure B.5). While the images from other datasets considered in this work are binary and contain only 2D objects, the CLEVR-ART images are colored and contain 3D objects with such real-world features as shades and reflections. Hence, the glimpsing behavior can be distracted by spurious salient locations such as edges of shades or regions of high reflection. Table 4 highlights that the GAP-Abstractor performs on par with the state-of-the-art model, Slot-Abstractor, being marginally better in one task and marginally worse in the other. However, Slot-Abstractor was pre-trained on all shapes from CLEVR-ART being thereby potentially exposed to objects from both train and test sets. This is in contrast to our models, which were not pre-trained and were exposed only to the objects from the train set. To compare our models with Slot-

Table 4: **Test accuracy for CLEVR-ART**. The accuracy results are averaged over 5 trained models. Prior art results are listed in the top section whereas results obtained from our evaluations are listed in the two bottom sections. The highest and the second-highest mean accuracy in each column is marked in **bold** and underlined, respectively. The models appended by * contain a module to segment the shapes that was pre-trained on images containing shapes from both the train and test datasets.

| Model | Task accuracy [%] | |
|---|---|---|
| | RMTS | ID |
| GAMR | 70.4 ± 5.8 | 74.2 ± 4.0 |
| OCRA* | 93.3 ± 1.0 | 77.1 ± 0.7 |
| Slot-Abstractor* | **96.3 ± 0.5** | 91.6 ± 0.2 |
| Transformer | 64.9 ± 3.4 | 52.2 ± 4.3 |
| GAP-Transformer | 76.4 ± 4.3 | 64.7 ± 5.6 |
| Abstractor | 77.8 ± 12.2 | 82.2 ± 3.8 |
| GAP-Abstractor | 95.9 ± 1.4 | **93.4 ± 1.3** |

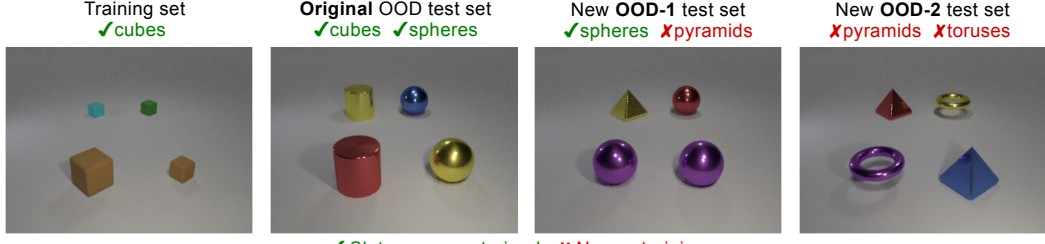

Figure 6: **Extension to CLEVR-ART for more thorough evaluation of OOD generalization**. Examples for the RMTS task are shown here. The two new OOD test sets contain objects on which neither of the models was pre-trained: OOD-1 test set contains 1 novel object (pyramid) and OOD-2 contains 2 novel objects (pyramid and torus).

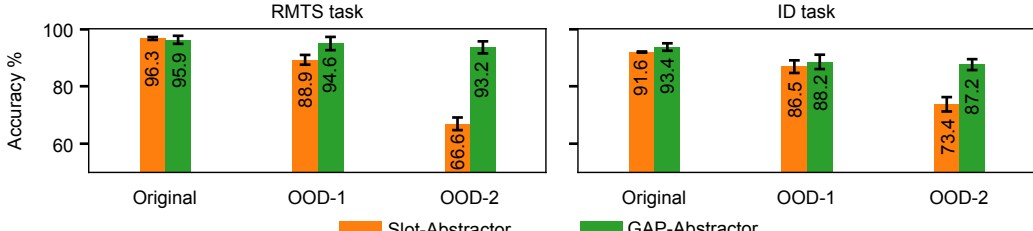

Figure 7: **Extended OOD evaluation on CLEVR-ART**. While the Slot-Abstractor was pre-trained on the original test set, the OOD-1 and OOD-2 test sets contain shapes on which neither of the evaluated models was pre-trained. The accuracy results are averaged over 5 trained models.

Abstractor on a more equal footing, we generated two additional OOD test sets shown in Figure 6. The new OOD-1 test set contains one novel type of shape – pyramid – on which the Slot-Abstractor was not pre-trained. Similarly, the new OOD-2 test set contains 2 such novel shapes: pyramids and toruses. The results are shown in Figure 7 demonstrating superior OOD generalization of our GAP-based model compared with Slot-Abstractor. In addition, we provide preliminary experiments with more realistic objects and more complex saliency map extraction in Appendix E.

## 6 DISCUSSION

Our results suggest that factorizing an image into its complementary "what" and "where" contents plays an essential role. The low-dimensional "where" content (glimpse locations) is crucial since it does not contain any visual information and, in turn, allows to learn relations that are agnostic to specific visual details. To capitalize on that, it is important to process the "where" content explicitly, disentangling it from its "what" complement. We implemented this by using the recently introduced relational cross-attention mechanism (employed by the Abstractor downstream architecture) where the "what" content is used to compute the attention scores for the processing of the "where" content. In contrast, implicitly mixing the "what" and the "where" contents, as in the case of the Transformer downstream architecture, weakens the generalization capabilities. This can be seen by comparing the performance between GAP-Transformer and GAP-Abstractor. To further support this, we provide supplementary results in Appendix D.2 showing the importance of using both the "what" and the "where" contents instead of just one of them for task solving.

Another important aspect of our model is that it processes only the most salient image regions while ignoring the unimportant ones. The inferior performance of models where the downstream architectures receive full information (i.e. all patches that constitute the image) suggests that supplying a superfluous amount of information may distract the models hindering the learning process. Additional results provided in Appendix D.3 elucidate this further. Specifically, we show that it is insufficient to naively reduce the amount of information provided to the downstream architecture, by discarding uninformative image parts. Instead, it has to be ensured that the supplied information is well structured. In the context of GAP, it means that the glimpse contents have to be centered around salient image regions. For example, the image patches of the glimpse contents should contain objects' edges in their central parts rather than somewhere in the periphery.

## 7 CONCLUSION

Drawing inspiration from human eye movements and theories of active vision, we proposed a system equipped with the novel Glimpse-based Active Perception (GAP). The system selectively glimpses at the most salient image parts and processes them in high resolution. The relational geometry between the corresponding glimpse locations complemented by the glimpse contents provides a basis for reasoning about image structure, i.e. relations between different image parts. We evaluated our approach on four visual reasoning benchmarks and achieved state-of-the-art performance in terms of sample efficiency and the ability to generalize to OOD visual inputs without any pre-training. Our approach allows for further extension by integrating more advanced components. Particularly, the saliency map extraction can be substantially enhanced by either extending our concept of error neurons or integrating more powerful components.

ACKNOWLEDGMENTS

The authors would like to thank Wolfgang Maass for the helpful comments and fruitful discussions.

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

## A    ABSTRACTOR

To describe the Abstractor-based downstream architecture and the way it is applied, we follow (Mondal et al., 2024). Specifically, given a sequence of glimpse contents $g = (g_1, g_2, \ldots, g_T)$ and their locations $x = (x_1, x_2, \ldots, x_T)$, the Abstractor computes a sequence of the corresponding relational representations $r$ over a series of $L$ layers. The initial relational representations $r_{l=0}$ correspond to the glimpse locations $x$. Each subsequent layer $l$ updates the relational representations using multi-head relational cross-attention between the glimpse contents $g$ (or their latent representations, if they are further pre-processed by a neural network):

$$\tilde{r}_h = \text{softmax}\left(\frac{(gW_q^h)^T(gW_k^h)}{\sqrt{d}}\right) r_{l-1}W_v^h, \tag{4}$$

$$r_l = \text{RCA}(g, r_{l-1}) = \text{concat}(\tilde{r}_{h=1}, \ldots, \tilde{r}_{h=H})W_o \tag{5}$$

where $W_q^h, W_k^h, W_v^h \in \mathbb{R}^{d \times d}$ ($d$ is the number of features of each head), are the linear projection matrices used by the $h^{\text{th}}$ head to generate queries, keys, and values respectively; $\tilde{r}_h$ is the result of relational cross-attention in the $h^{\text{th}}$ head; $W_o$ are the output weights through which the concatenated outputs of all $H$ heads are passed. Importantly, in RCA the queries and keys are produced based on glimpse contents, i.e. based on visual information, so that the resulting inner product in the softmax of Eq. 4 retrieves the visual relations disentangled from the specific visual features. At the same time, the values are generated from the glimpse locations that are stripped of any visual information. Hence, the overall processing in RCA is driven only by the relations between visual features (but not the visual features themselves) that modulate the processing of structural information (glimpse locations). Similar to (Mondal et al., 2024), in each layer, RCA was appended with feedforward networks and standard self-attention (Vaswani et al., 2017) as well as residual connections between each of those components (Figure A.1).

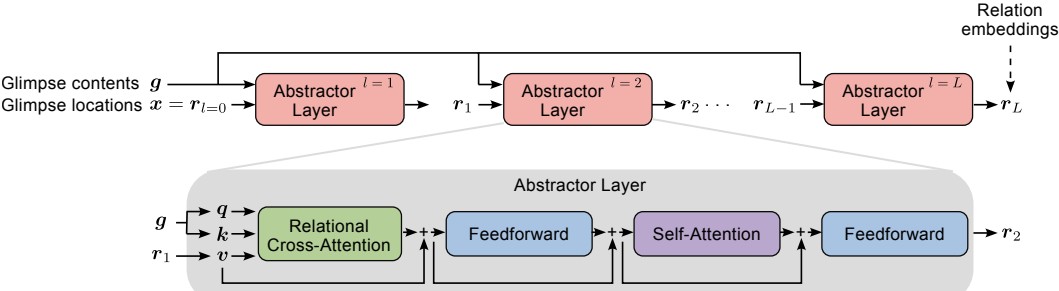

Figure A.1: **Abstractor downstream architecture**: consists of several layers each consisting of the relational cross-attention (RCA), feedforward networks, the standard self-attention (SA), and residual connections between those. The most important component, the RCA, differs from the SA in that the queries and keys are generated based on glimpse contents (or their visual features) whereas the values are generated based on the glimpse locations. The figure is adapted from (Mondal et al., 2024).

## B    DATASET DESCRIPTIONS

### B.1    SVRT

The Synthetic Visual Reasoning Test (SVRT) dataset, introduced by (Fleuret et al., 2011), comprises 23 binary classification tasks. Each task involves a set of synthetic 2D shapes with an underlying relation between them. Following (Vaishnav & Serre, 2023), these tasks are divided into two families: those defined by same/different relations (SD) and those defined by spatial relations (SR). SD tasks: 1, 5, 6, 7, 13, 16, 17, 19, 20, 21, 22; SR tasks: 2, 3, 4, 8, 9, 10, 11, 12, 14, 15, 18, 23. The main challenge with these tasks lies in training with very few samples, thus testing the models' inductive biases for learning the relational aspects of an image. The most commonly used dataset splits

consider 500 or 1000 examples per task. The validation and test sets contain 4k and 40k samples respectively.

## B.2 SVRT #1-OOD

This dataset is introduced by (Puebla & Bowers, 2022) and is built around the same-different task #1 from SVRT, in which the model has to make a binary decision whether two shapes in the image are the same up to a translation. While this dataset contains the same training images as in the original SVRT dataset, it introduces 13 test sets with novel OOD shapes that significantly differ from those in the train set (Figure B.2). The dataset contains 28k and 5.6k images for training and validation respectively. Each OOD test set contains 11.2k images.

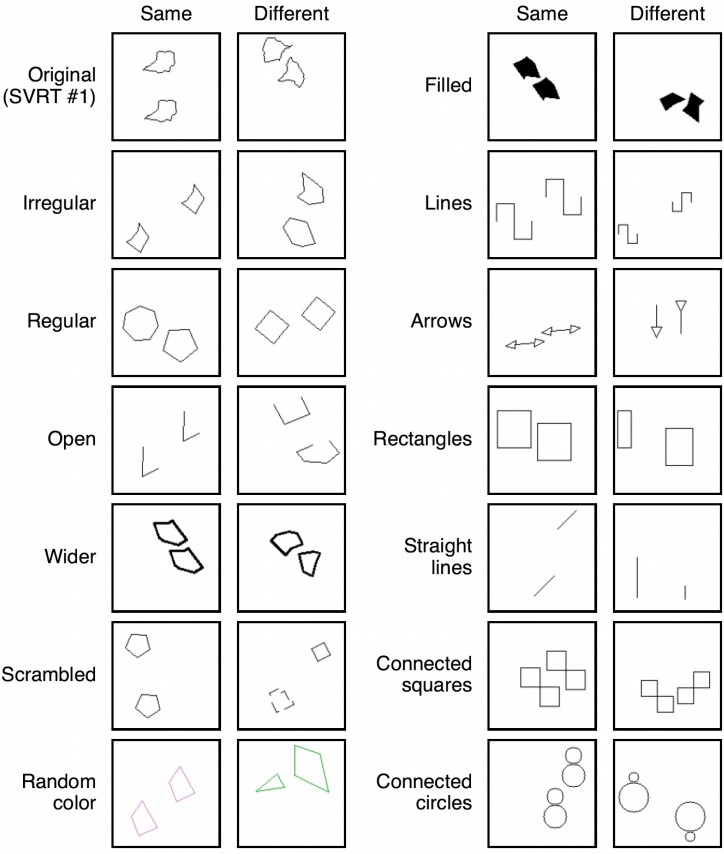

Figure B.2: **SVRT #1-OOD** dataset. The goal is to determine whether 2 shapes are the same (up to a translation) or different. The train set consists of images from the original SVRT dataset. The test set includes 13 subsets each containing a different type of out-of-distribution shapes. Taken from (Puebla & Bowers, 2023).

## B.3 ART

The ART (Abstract Reasoning Tasks) dataset, proposed by (Webb et al., 2021) comprises four visual reasoning tasks each having a distinct underlying abstract rule to be detected: same/different (SD), relational-match-to-sample (RMTS), distribution-of-3 (Dist-3) and identity rules (ID). Figure B.4 illustrates each of the tasks. The dataset was constructed using 100 Unicode character objects, and it includes generalization regimes of varying difficulty based on the number of unique objects used during training. Following (Mondal et al., 2024), we focused on the most challenging generalization regime, where the train set involves problems created from 5 out of the 100 possible objects, while the test problems use the remaining 95 objects. This setup tests systematic generalization, requiring abstract rule learning from a small set of examples with minimal perceptual overlap between train

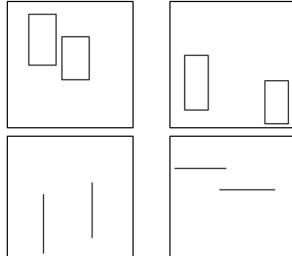

Figure B.3: **Difficult cases of SVRT #1-OOD**: The shown images are from the 'straight_lines' and 'rectangles' OOD test sets. The images contain different shapes that might be difficult to distinguish even for humans.

and test sets. SD, RMTS, Dist-3, and ID tasks contain 40, 480, 360, and 8640 training examples respectively while all of them contain 20K and 10K validation and test examples respectively.

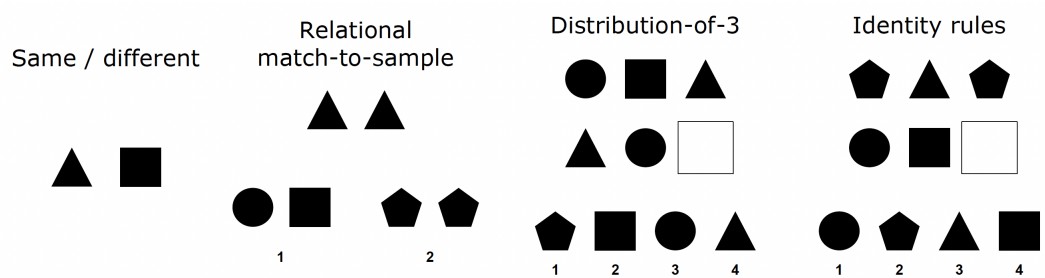

Figure B.4: **Abstract Reasoning Tasks (ART) dataset** consists of 4 tasks. In the same/different task, the model has to determine whether an image contains 2 the same (up to a translation only) shapes. In the relational-match-to-sample task, the model has to select a pair of objects out of two pair candidates presented in the bottom row that contains objects in the same relation (either the 'same' or 'different') as the objects in the upper row. In the distribution-of-3 task, the model has to pick an object so that the lower row contains 3 objects that comprise the same set of objects as that of the upper row. In the identity rules task, the model has to select an object so that the resulting lower row contains objects that follow the same abstract pattern (ABA, ABB or AAA) as the objects in the upper row. Each instance of each task (except the same/different task) was presented with multiple images each containing a separate candidate object (or object pair) so that the model has to pick the one with the correct pair. Taken from (Webb et al., 2021).

## B.4 CLEVR-ART

The CLEVR-ART dataset Figure B.5, proposed by (Webb et al., 2024a), utilizes photorealistic synthetic 3D shapes from CLEVR (Johnson et al., 2017). It includes two visual reasoning tasks from ART: relational-match-to-sample and identity rules. The train set consists of images created using small and medium-sized rubber cubes in four colors (cyan, brown, green, and gray). In contrast, the test set features images generated from large-sized metal spheres and cylinders in four different colors (yellow, purple, blue, and red). Hence, the object features in the train and test sets are entirely distinct, challenging the systematic generalization of learned abstract rules to previously unseen object characteristics.

## C EXPERIMENTAL DETAILS

For all tasks, the images were resized to $128 \times 128$ and the pixels were normalized to the range $[0, 1]$. For SVRT tasks, random horizontal and vertical flips were applied during the training following (Vaishnav & Serre, 2023).

Relational match-to-sample

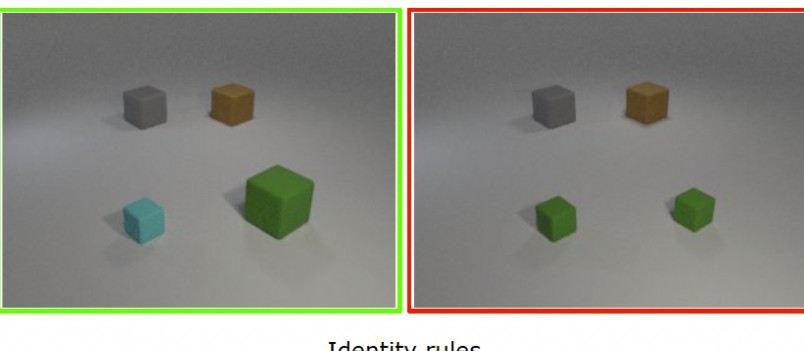

Identity rules

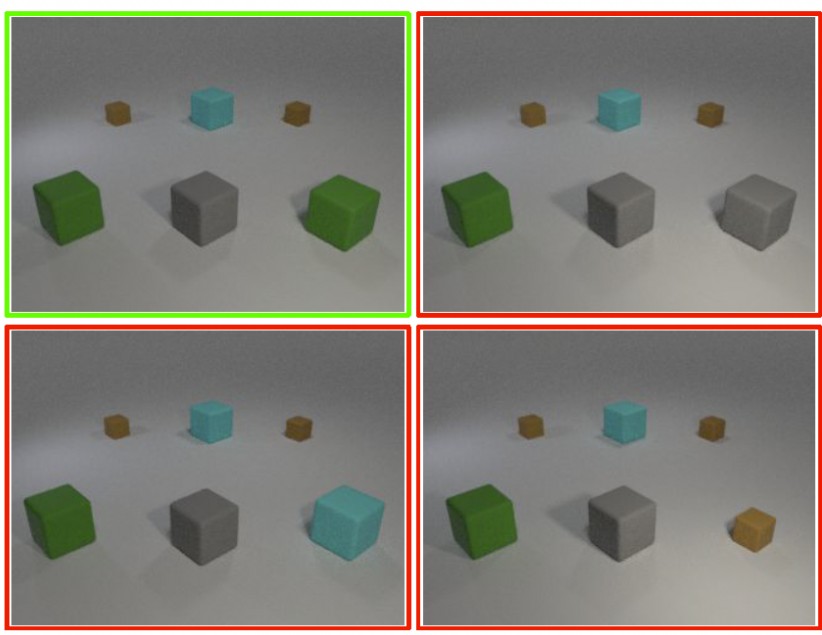

Figure B.5: **CLEVR-ART dataset** consists of two tasks from ART dataset. Relational match-to-sample task: example problem involving the relation of difference. The correct answer choice (left image) contains pairs of different objects in the back and the front rows. The incorrect answer choice (right image) contains a pair of different objects in the front row but the same objects in the back row. Identity rules: example problem involving ABA rule. The correct answer choice (top left image) involves ABA rule in both back row and front row of objects. The object in the front row on the right in the other three images (incorrect choices) violates this rule. Taken from (Webb et al., 2024a).

At each location of the saliency map, the corresponding error neuron compares a small (central) image patch of size $5 \times 5$ that is in its receptive field with 8 surrounding patches of the same size. The surrounding patches are obtained by sliding a window from the neuron's receptive field with the stride of 1 in 8 cardinal directions so that the surrounding patches have a high overlap with the central patch. The aggregation operator takes the minimal difference between the central patch and each of the surrounding patches.

The number of iterations $T$ in the glimpsing process is set to 15 for SVRT (and SVRT #1-OOD), 20 for SD and RMTS tasks from ART, and 35 for all other tasks from ART and CLEVR-ART. For all datasets, the IoR mechanism used the hard mask (with radius of 5 pixels) except for CLEVR-ART where its soft version (with $\epsilon = 450$) was used. For all datasets, the multi-scale glimpse sensor provided glimpses at 3 scales each of which was of size $15 \times 15$, where the first scale corresponded

to the original resolution and the other two downsampled the regions of size $30 \times 30$ and $45 \times 45$. For the log-polar sensor, we use OpenCV (Bradski, 2000) formulation and implementation of the log-polar sampling setting the radius of the log-polar grid was set to 48. The glimpse size was set to 21 for SVRT (and SVRT #1-OOD) and ART tasks and to 15 for CLEVR-ART.

In our glimpse-based models, we use a small CNN to pre-process each glimpse content before feeding it to the downstream architecture. For glimpse contents produced by the multi-scale sensor, we use a 7-layer CNN where each layer has the kernel size of $3 \times 3$, no padding and a ReLU activation function. For glimpse contents produced by the log-polar sensor, we use almost the same CNN but with 6 layers where the first 4 layers have the kernel size of $5 \times 5$. The number of channels per layer is the same in both cases being 9 for SVRT (and SVRT #1-OOD) and 64 for the rest of the datasets. Each glimpse location was re-scaled to be in range $[-1, 1]$ and was pre-processed by an MLP with with two layers of size 32 and 64. As in (Mondal et al., 2024), we applied temporal context normalization (TCN) (Webb et al., 2020) on both sequences of pre-processed glimpse contents and locations. TCN normalizes a sequence along its temporal dimension.

For all datasets, the Abstractor module was instantiated with 8 attention 64-dimensional heads, 24 layers, and 2-layered MLPs with the hidden size of 256. The 50% dropout for the MLPs and attention is used only for SVRT #1-OOD. To generate the final output for the task, we took the mean of the sequence of final representations $r_L$ and passed it to a single linear unit.

For the training, we used ADAM optimizer (Kingma & Ba, 2015) with the learning rate of $1e - 4$, batch size of 64. The whole model was trained end-to-end with the binary cross-entropy loss.

# D    ABLATIONS AND ADDITIONAL RESULTS

Table D.1: Test accuracy (in %) for SVRT dataset using different glimpse sensors. The accuracy results are averaged over SD and SR tasks obtained from the best trained model.

| Model | Glimpse sensor | SD tasks | | SR tasks | |
|---|---|---|---|---|---|
| | | Dataset size 1000 | Dataset size 500 | Dataset size 1000 | Dataset size 500 |
| GAP-Transformer | multi-scale | 80.8 ± 13.5 | 68.0 ± 14.3 | 97.9 ± 2.5 | 96.5 ± 3.1 |
| GAP-Transformer | log-polar | **83.3 ± 13.2** | **78.4 ± 13.5** | **98.2 ± 2.2** | **97.3 ± 2.7** |
| GAP-Abstractor | multi-scale | **93.1 ± 13.1** | 90.5 ± 14.5 | **98.5 ± 1.9** | 96.6 ± 2.1 |
| GAP-Abstractor | log-polar | 92.5 ± 12.3 | **90.6 ± 13.3** | 98.2 ± 2.6 | **97.0 ± 3.4** |

Table D.2: OOD performance (in %) for test sets from SVRT #1-OOD using different glimpse sensors. The accuracy results are averaged over all OOD test sets for 10 trained models.

| Model | Glimpse sensor | Accuracy (averaged over runs and OOD datasets) |
|---|---|---|
| GAP-Transformer | multi-scale | **76.3 ± 18.4** |
| GAP-Transformer | log-polar | 74.7 ± 16.0 |
| GAP-Abstractor | multi-scale | 84.4 ± 15.3 |
| GAP-Abstractor | log-polar | **89.6 ± 8.7** |

Table D.3: Accuracy (in %) for tasks from the ART dataset using different glimpse sensors. The accuracy results are averaged over 10 trained models.

| Model | Glimpse sensor | Task | | | |
|---|---|---|---|---|---|
| | | SD | RMTS | Dist3 | ID |
| GAP-Transformer | multi-scale | **76.4 ± 13.7** | **62.2 ± 7.3** | 50.7 ± 6.1 | 55.5 ± 5.8 |
| GAP-Transformer | log-polar | 64.3 ± 9.1 | 52.6 ± 0.8 | **51.3 ± 13.7** | **57.0 ± 8.8** |
| GAP-Abstractor | multi-scale | 95.8 ± 0.8 | 95.3 ± 2.0 | 93.9 ± 1.3 | 91.1 ± 4.5 |
| GAP-Abstractor | log-polar | **97.7 ± 2.0** | **96.3 ± 1.0** | **98.4 ± 0.5** | **96.8 ± 1.8** |

Table D.4: Accuracy (in %) for tasks from the CLEVR-ART dataset using different glimpse sensors. The accuracy results are averaged over 5 trained models.

| Model | Glimpse sensor | Task | |
| --- | --- | --- | --- |
| | | RMTS | ID |
| GAP-Transformer | multi-scale | **76.4 ± 4.3** | **64.7 ± 5.6** |
| GAP-Transformer | log-polar | 70.0 ± 3.5 | 63.0 ± 8.0 |
| GAP-Abstractor | multi-scale | **95.9 ± 1.4** | **93.4 ± 1.3** |
| GAP-Abstractor | log-polar | 94.6 ± 2.0 | 92.5 ± 1.5 |

## D.1 PERFORMANCE DIFFERENCES BETWEEN GLIMPSE SENSORS

We evaluated two different glimpse sensors for each of the four datasets considered in this work, see Table D.1- D.4 and Figure D.6. For the Abstractor-based downstream architecture, we observe comparable performance for both sensors in most cases. The most significant performance difference is observed for SVRT #1-OOD and two tasks from ART dataset (Table D.2-D.3) where the log-polar sensor achieves around 5% better accuracy. For the Transformer-based downstream architecture, the performance difference between two glimpse sensors averaged over all tasks is around 5%. However, there are tasks where the multi-scale sensor results in around 10% worse accuracy, see Table D.1, and 10% better accuracy, see Table D.3. Overall, based on the performance data, there is no clear quantitative trend favoring either of the two glimpse sensors. We ascribe this to the qualitative advantages and disadvantages of both sensors. In particular, the log-polar sensor benefits from the properties of the log-polar space where the rotation and scaling in the Cartesian space become translations, see e.g. (Javier Traver & Bernardino, 2010). However, the warping into log-polar space may make it difficult to capture information about image parts that are far from the glimpse locations. The multi-scale sensor, in contrast, captures the distant information more faithfully using downsampled image patches of different sizes. The disadvantage of this, however, is that the resulting glimpse content contains additional dimensions for the different scales.

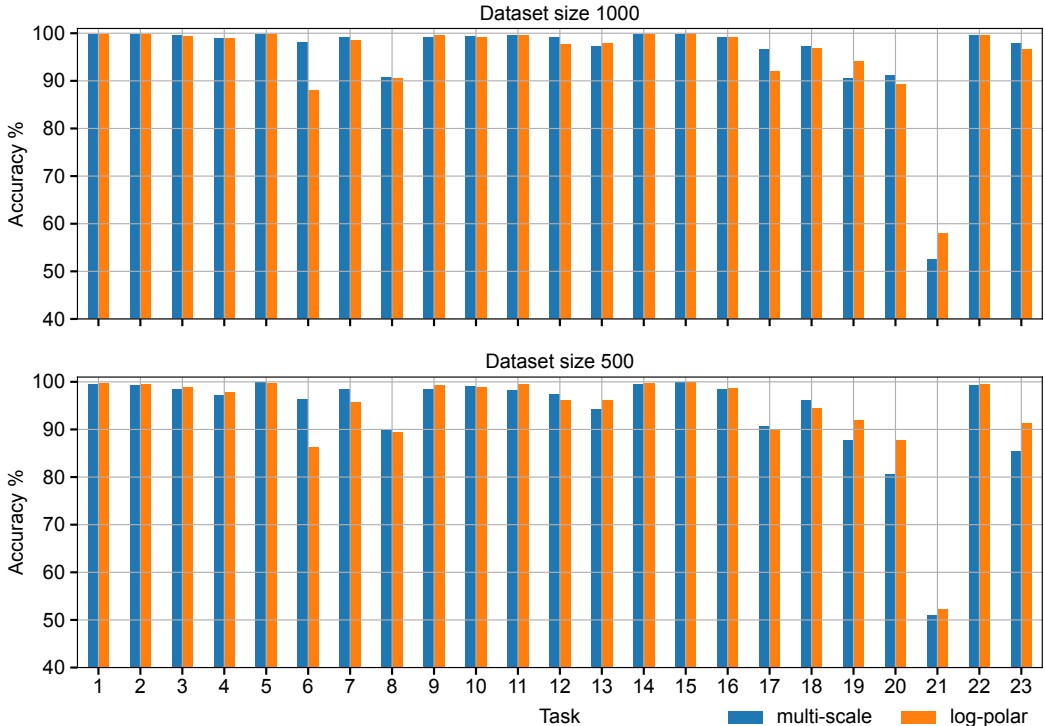

Figure D.6: Test accuracy for each of 23 SVRT tasks using different glimpse sensors for the GAP-Abstractor model.

Table D.5: Sample efficiency on the SVRT dataset for 11 same-different (SD) tasks and 12 spatial relations (SR) tasks depending on the size of the training dataset. The results report the test accuracy [%] for the best trained models, the standard deviation is computed over SD and SR tasks. Only our models are listed. The prior art models are shown in Fig. D.7

| Dataset size | SD tasks | | | |
| --- | --- | --- | --- | --- |
| | GAP-Abstractor | GAP-Transformer | Abstractor | Transformer |
| 10000 | 96.2 ± 6.7 | 95.3 ± 6.2 | 52.2 ± 4.8 | 57.4 ± 14.5 |
| 5000 | 95.4 ± 8.1 | 92.7 ± 9.8 | 51.7 ± 3.9 | 56.5 ± 14.2 |
| 1000 | 93.1 ± 13.1 | 82.5 ± 15.1 | 51.2 ± 2.1 | 51.9 ± 2.7 |
| 500 | 90.5 ± 14.5 | 76.0 ± 13.4 | 51.0 ± 1.9 | 51.6 ± 1.7 |
| 250 | 86.8 ± 16.0 | 69.4 ± 15.3 | 50.9 ± 1.5 | 51.3 ± 1.2 |
| 100 | 74.3 ± 17.2 | 58.7 ± 6.2 | 50.6 ± 0.9 | 51.0 ± 0.8 |

| Dataset size | SR tasks | | | |
| --- | --- | --- | --- | --- |
| | GAP-Abstractor | GAP-Transformer | Abstractor | Transformer |
| 10000 | 99.7 ± 0.7 | 99.7 ± 0.5 | 93.5 ± 13.9 | 82.9 ± 19.2 |
| 5000 | 99.6 ± 0.5 | 99.6 ± 0.8 | 86.5 ± 17.3 | 75.7 ± 20.3 |
| 1000 | 98.5 ± 1.9 | 98.2 ± 2.2 | 60.9 ± 14.3 | 64.8 ± 13.7 |
| 500 | 96.6 ± 1.2 | 97.2 ± 2.5 | 57.0 ± 11.0 | 59.9 ± 11.0 |
| 250 | 93.4 ± 6.3 | 95.1 ± 3.5 | 52.4 ± 3.8 | 58.1 ± 8.8 |
| 100 | 86.2 ± 13.4 | 86.9 ± 11.2 | 52.2 ± 3.3 | 54.5 ± 5.1 |

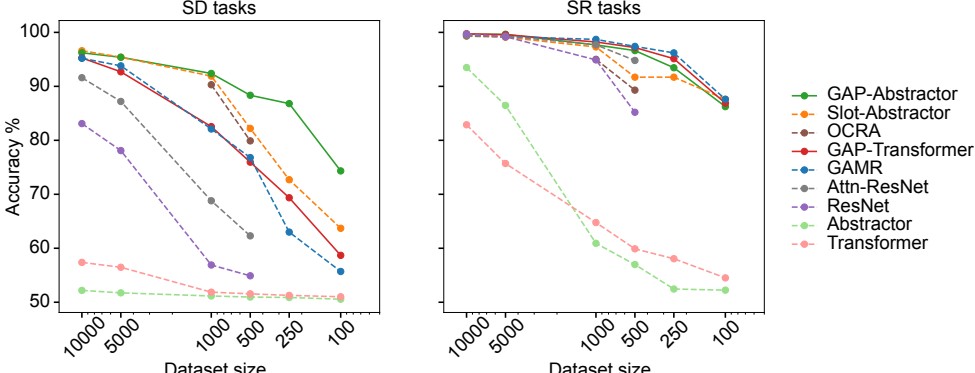

Figure D.7: Sample efficiency on the SVRT dataset for 11 same-different (SD) tasks and 12 spatial relations (SR) tasks depending on the size of the training dataset. The results report the test accuracy [%] for the best trained models, the standard deviation is computed over SD and SR tasks.

## D.2 EVALUATING IMPORTANCE OF GLIMPSE CONTENTS AND GLIMPSE LOCATIONS

To measure the extent to which glimpse contents ("what") and glimpse locations ("where") are important, we trained our models passing only one of those two sequences to their downstream architectures. The training was done on SD tasks from SVRT dataset using 1000 training samples. For the Transformer-based downstream architecture, the input sequence consisted of either glimpse contents or glimpse locations, instead of the concatenation thereof. For Abstractor-based downstream architecture, we used a fixed set of trainable embeddings from which the RCA produced values while the keys and the queries were produced from either of the aforementioned input sequences. As it can be seen from Table D.6, the combination of glimpse contents and glimpse locations yields the highest performance compared to cases where only one of those sequences is used.

## D.3 EVALUATING THE ROLE OF PROCESSING ONLY THE RELEVANT INFORMATION

The superior performance of our GAP-based models compared to the ablated downstream architectures that process entire images indicates the importance of processing only the relevant information. The relevance of information in the context of GAP entails two features explained below.

Table D.6: Test accuracy [%] averaged over SVRT SD tasks (11 in total) for models trained with 1000 samples depending on information passed to the downstream architecture.

| Model | Inputs to the downstream architecture | | |
|---|---|---|---|
| | glimpse contents | glimpse locations | glimpse contents and locations |
| GAP-Abstractor | 81.0 ± 21.9 | 66.3 ± 14.2 | 93.1 ± 13.1 |
| GAP-Transformer | 77.4 ± 12.1 | 72.8 ± 17.5 | 83.3 ± 13.2 |

Table D.7: Test accuracy [%] averaged over SVRT SD tasks (11 in total) for models trained with 1000 samples depending on information passed to the downstream architecture. See Section D.3 for more information.

| Downstream architecture | All patches (#patches=12996) | ViT patches (#patches=64) | GAP-regular (#patches=15) | GAP (#patches=15) |
|---|---|---|---|---|
| Abstractor | 50.2 ± 1.3 | 51.3 ± 2.5 | 58.6 ± 14.8 | 93.1 ± 13.1 |
| Transformer | 50.9 ± 1.9 | 51.9 ± 2.6 | 53.8 ± 6.1 | 83.3 ± 13.2 |

First, GAP discards all uninformative parts of the image providing thereby much less information to the downstream architecture than is available. This, in turn, improves the learning process preventing the models from being distracted by superfluous information. To elucidate the importance of this feature, we show the comparison of our GAP approach to the downstream architectures that process i) all image patches obtained by the sliding window approach with the stride of one (such patches almost entirely overlap and include glimpse contents obtainable by GAP); ii) all non-overlapping image patches obtained in ViT-like fashion based on the regular grid. The cases i) and ii) are referred to as *all patches* and *ViT patches* respectively.

Second, each glimpse content is focused meaning that it is centered around a salient image region. This is, for example, in stark contrast to the patching process taking place in ViTs where the image is split into non-overlapping patches based on a regular grid. In that case, even if there were a method to determine only a subset of the most relevant patches, those patches would not necessarily contain the best portions of salient regions. To show the importance of focusing, we compare GAP to its modification where glimpse locations are tweaked to be in a set of "coarse" locations defined by the ViT's regular grid. Importantly, this set of locations is only a subset of all possible image locations that the GAP can glimpse at. The resulting glimpse contents are obtained from the corresponding tweaked glimpse locations. We refer to this modification of GAP as *GAP-regular*.

All models described above were trained on SVRT SD tasks (11 tasks in total) using 1000 training samples. The size of the patches was the same as the size of the glimpse contents ($15 \times 15$) and the number of glimpses for GAP and GAP-regular was set to 15. The total number of patches obtained in the cases of all patches and ViT patches is 12996 and 64 respectively.

Table D.7 shows the corresponding results. It can be seen that neither of the two aforementioned features alone is sufficient to solve the tasks. Specifically, providing all information available in face of either all patches or ViT-patches makes the downstream architecture not able to solve any task. The same can be seen in the case of providing less information but not well focused on salient regions (although the performance of GAP-regular is slightly above the 50% chance level). In other words, it is important to process only the relevant information by providing only a subset of available information and making this subset contain information focused on the most salient image parts.

## E    EXPERIMENT WITH MORE REALISTIC OBJECTS AND ALTERNATIVE SALIENCY MAPS

We evaluated our model on more realistic objects. Additionally, we show how different saliency maps may improve the model's performance in this case.

In particular, we used the objects from the Super-CLEVR dataset (Li et al., 2023), including cars, planes and motorbikes, to generate a new test set for the Identity Rules task from the CLEVR-ART dataset. The task is to determine whether three objects in the top row are arranged under the same abstract rule as the objects the bottom row. For example, objects "bus", "car" and "bus" in one row

are arranged under the abstract pattern ABA just as objects "car", "bus" and "car. We considered four models trained on the original CLEVR-ART train set that consists only of simple cubes (Figure 6), and evaluated them on the new dataset, referred to as Super-CLEVR test set. The example images from the Super-CLEVR test set are shown in Figure E.8.

Simultaneously, we explored using different saliency maps. The first model is our GAP-Abstractor model with the original error neurons applied on the raw image to extract the saliency map, we refer to this model here as GAP-Abstractor (EN). The second model, referred to as GAP-Abstractor (CNN+EN), is a modification of GAP-Abstractor that extracts the saliency map by applying the error neurons on the feature maps computed by the first layer of ResNet-18 that was pre-trained on ImageNet. Computing the saliency map from the feature maps should make the glimpsing behavior less distracted by spurious salient locations, such as edges of shades or regions of high reflection. The third model, referred to as GAP-Abstractor (IK), employs an alternative way to extract the saliency maps using a more complex bio-plausible model proposed by Itti et al. (1998). The example saliency maps are shown in Figure E.8. Table E.8 shows the corresponding results, where Slot-Abstractor is the best performing prior-art model providing the baseline.

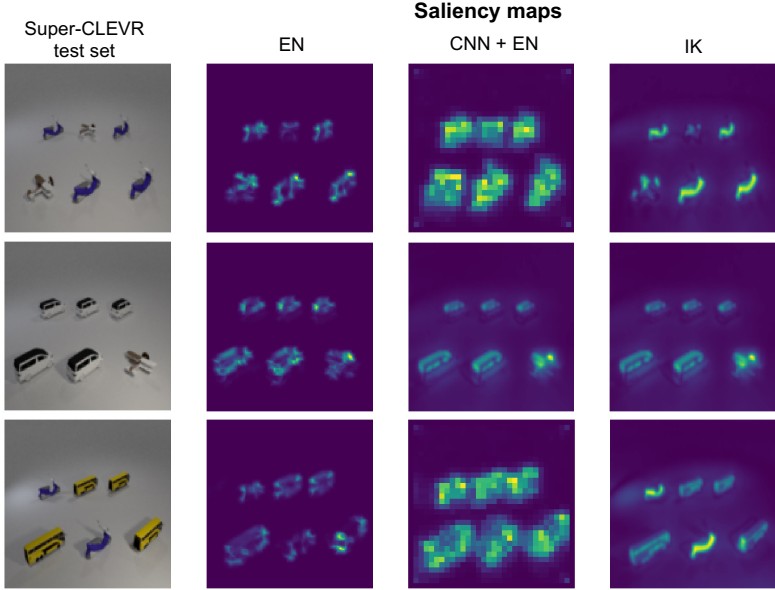

Figure E.8: Different saliency maps for images with more realistic objects. The images were generated using objects from the Super-CLEVR dataset for the ID task defined by the CLEVR-ART dataset.

Table E.8: Accuracy [%] on ID task from CLEVR-ART on the original test set and the OOD test with more realistic objects from Super-CLEVR. The accuracy results are averaged over 5 trained models. The highest and the second-highest mean accuracy in each column is marked in bold and underlined, respectively.

| Model | original test set | Super-CLEVR test set |
|---|---|---|
| Slot-Abstractor (prior art) | 91.61 ± 0.2 | 24.1 ± 0.8 |
| GAP-Abstractor (EN) | **93.4 ± 1.3** | 72.3 ± 2.3 |
| GAP-Abstractor (CNN+EN) | 92.7 ± 1.8 | 80.2 ± 1.9 |
| GAP-Abstractor (IK) | 90.2 ± 2.1 | **82.1 ± 2.2** |

As it can be seen, Slot-Abstractor fails completely on the unseen Super-CLEVR objects while the performance of our GAP-Abstractor does not degrade so severely. Moreover, we see that GAP-Abstractor (CNN+EN) performs better on Super-CLEVR test set while performing comparably on the original test set. Therefore, applying the error neurons on top of features maps rather than

on top of the raw images results in a saliency map that can better steer the glimpsing behavior. Lastly, we see that GAP-Abstractor (IK) with an alternative bio-plausible way to extract the saliency map reaches the best performance on the Super-CLEVR test set while being slightly worse on the original test set. A possible reason for this difference is that this saliency map extractor was designed to handle real-world scenes that are of much higher visual complexity compared to simple shapes. Overall, these results indicate a possible direction of enhancing our approach for visual reasoning by integrating more powerful saliency maps.

