# OpenReview forum: "Mind the GAP: Glimpse-based Active Perception improves generalization and sample efficiency of visual reasoning"
_ICLR.cc/2025/Conference — ICLR 2025 Poster_

### Official Review · Reviewer_KWe3 · 2024-11-01

**Soundness:** 3
**Presentation:** 3
**Contribution:** 2
**Rating:** 6
**Confidence:** 3

**Summary:**

This paper discusses how to improve the performance of visual reasoning by simulating the active perception of humans, especially when dealing with unknown objects. By expressing the relationship between different image parts with the position generated by glimpse-based active perception and the visual parts around them, this approach reaches a better sampling efficiency and generalization performance.

**Strengths:**

1. It is really inspiring to simulate human's active perception to improve the performance of visual reasoning, which is a contribution to the community of visual reasoning.
2. The approach achieves better sampling efficiency and generalization performance.
3. The authors provide extensive experiment results and analysis.

**Weaknesses:**

1. For the discussion of generalization ability, the paper focuses on the OOD data of the specific benchmarks. However, what we hope to achieve is the ability that models can deal with totally different tasks like humans, although it is really though. This paper lacks the analysis of it.
2. Recently, many LLM-based approaches are gaining higher performance on the reasoning tasks like VQAs. More analysis of the relationship between pure abstract reasoning like this approach and LLM-based reasoning is desirable.

**Questions:**

1. What concerns me most is the generalization ability of this approach, although it is inspiring and fancy. Could you give me some clues that this approach can deal with tasks across different domains to reach the authentic reasoning ability?
2. Many people assume that LLM-based reasoning is closest to the generalization due to its rich knowledge. I suppose this approach would also be helpful to the LLM community. Would you conduct some analysis on the relationship between this approach and LLM-based reasoning?

---

> ### Author Response · Authors · 2024-11-22
>
> We thank the Reviewer for constructive comments. Below we have prepared detailed responses to the points raised in the Weaknesses and Questions sections.
>
> > **Weakness**: For the discussion of generalization ability, the paper focuses on the OOD data of the specific benchmarks. However, what we hope to achieve is the ability that models can deal with totally different tasks like humans, although it is really though. This paper lacks the analysis of it
>
> &
>
> > **Question**: What concerns me most is the generalization ability of this approach, although it is inspiring and fancy. Could you give me some clues that this approach can deal with tasks across different domains to reach the authentic reasoning ability?
>
> Inspired by cognitive theories such as (Summerfield et al., 2020), our approach assumes the key role of action in information processing. According to those theories, actions not only allow to gather information about task-relevant parts of the environment, but they also provide information about relations between those parts. In other words, actions allow to “connect the dots” where the “dots” are the single pieces of information about the environment obtained after taking each action.
> We believe that actions have two important properties that make them important for generalization across different sensory domains.
> Firstly, actions tend to have low-dimensional representations, be that information about eye movements to process the visual information or information about finger movements for processing tactile information. Secondly, information about actions has no overlap with its sensory counterpart meaning that the same set of actions can describe the same spatial or abstract relation between completely different parts of the sensory environment.
> Hence, given the fact that actions control not only our visual sensors but also sensors of other senses (e.g. haptic, auditory), these two properties suggest that actions underpin reasoning capabilities in various domains.
>
> References
> - Summerfield, C., Luyckx, F., & Sheahan, H. (2020). Structure learning and the posterior parietal cortex. Progress in Neurobiology.

---

> > ### Author Response · Authors · 2024-11-22
> >
> > > **Weakness**: Recently, many LLM-based approaches are gaining higher performance on the reasoning tasks like VQAs. More analysis of the relationship between pure abstract reasoning like this approach and LLM-based reasoning is desirable.
> >
> > &
> >
> > > **Question**: Many people assume that LLM-based reasoning is closest to the generalization due to its rich knowledge. I suppose this approach would also be helpful to the LLM community. Would you conduct some analysis on the relationship between this approach and LLM-based reasoning?
> >
> > Thank you for these suggestions. To address this aspect, we conducted an experiment described in the general comment to all Reviewers “**Study on LLMs’ capabilities of solving same-different task**”

---

> ### Comment · Reviewer_KWe3 · 2024-11-24
>
> Yes, I understand your motivation. Starting from cognitive theories, this glimpse-based approach has theoretical potential for generalization. However, my concern is whether this method demonstrates generalization capabilities in experimental applications across other domains. Considering time constraints, I'm not requiring complete experimental results. Combining VLM's knowledge and this method should theoretically enable cross-domain generalization. I'm curious if there are any ways to **integrate this mechanism into VLM**. I suppose this would make this interesting method more applicable in the current VLM era.

---

> > ### Author Response · Authors · 2024-11-28
> >
> > We thank the Reviewer for this important question, and we agree that integrating our glimpse-based approach into VLMs is definitely interesting. We preliminarily explored two ways of integrating GAP into VLMs and evaluated them on the same-different task from the SVRT dataset. In this task, it has to be determined whether two shapes are the same up to translation.
> >
> > The first approach is to modify the vision encoding of VLMs, which typically comprises a Vision Transformer (ViT). In particular, our GAP approach could be integrated within the ViT such that, instead of a regular grid of patches from the entire image, the model would receive the relevant focused pieces of information in form of glimpse contents and locations. This approach is very similar to the analysis in Section 5 of our manuscript, where we demonstrated that the GAP-Transformer, e.g., a ViT combined with our glimpse-based method, shows improved OOD generalization compared to the plain ViT.  Thus, combining GAP with VLMs may also improve their generalization capabilities.
> >
> > We conducted preliminary experiments to explore the described idea using Pixtral, one of the SOTA VLMs. Although we could successfully incorporate our GAP approach into the processing pipeline, we could not observe any performance improvements. Trying to identify the reason, we created a simple prompt asking to compare only two image patches that corresponded to different parts of the shapes from SVRT. Surprisingly, Pixtral gave wrong answers being thereby unable to compare image patches containing simple parts of the shapes. This indicates that the features learned by VLM’s Transformer cannot faithfully represent the visual information inside single image patches. Hence, to successfully integrate our GAP approach into VLMs, the re-training of the vision encoding part is necessary. Given the short time frame and the size of Pixtral’s vision encoder, ~300M parameters, we could not conduct the corresponding experiment.
> >
> > The second approach is to keep the VLM unchanged but to modify the image that we provide to the VLM using the glimpse information from our GAP. We conducted an initial exploration where we marked the areas around glimpse locations in the input image with red squares. We envisioned that this incentivizes the VLM to analyze the important areas identified by GAP more carefully, treating them as regions of interest. Indeed, we could observe some accuracy improvement. For example, for the prompt
> >
> > _“Are the two shapes in the image the same? Following your explanation, please conclude your answer with the final 'yes' or 'no’.”_,
> >
> > Pixtral achieved 57% accuracy. And for the modified prompt
> >
> > _“Are the two shapes in the image the same? **The important parts of the shapes are within the red areas.** Following your explanation, please conclude your answer with the final 'yes' or 'no’.”_
> >
> > accompanied by the modified image with emphasized glimpse locations Pixtral achieved 68% accuracy. However, we noticed that the performance of the VLM highly depends on the prompt. Specifically, we tried various other prompts and observed that in some cases there was no performance difference between prompts with or without the GAP information. One such prompt is, for example,
> >
> > _“Is the second shape in the image an identical copy of the first shape just drawn at a different location? Please answer "yes" or "no".”_
> >
> > Overall, our second approach provided some initial indication of the usefulness of enhancing the input image with our GAP information, but the variation in the VLM performance for different prompts convinced us that the more promising path would be our first approach, where we integrate the GAP into the vision part of the VLM through re-training.

---

> > > ### Comment · Reviewer_KWe3 · 2024-11-29
> > >
> > > Thanks for your efforts in conducting an in-depth analysis of the two approaches to integrating the glimpse-based mechanism into VLM. I share the idea that integrating the GAP into the vision part of the VLM through re-training is more promising and paves the way for the improvement of current VLMs. I understand the time limitation of conducting the re-training experiments. The analysis has addressed my concern, so I will raise my score.

---

### Official Review · Reviewer_8myJ · 2024-11-01

**Soundness:** 3
**Presentation:** 3
**Contribution:** 2
**Rating:** 6
**Confidence:** 4

**Summary:**

The paper presents an architecture that performs visual perception based on local glimpses. It uses a saliency map to determine glimpse positions, and then feeds an encoding of the appearance and of the location of the glimpses to an existing downstream architecture - a Transformer or the recently proposed Abstractor. The glimpse extraction is hardcoded while the downstream architecture can be trained. The architecture is evaluated on reasoning tasks that rely on local structure and spatial relations.

**Strengths:**

The paper explores an architecture that make explicit the positions and not just appearance of local image regions. These types of architecture are interesting because of the potential capabilities they can enable, and because of their relation to human vision. The paper is well written and fairly easy to follow. The proposed architecture is very simple.

**Weaknesses:**

The proposed architecture can be viewed as a local feature extractor that encodes local image regions and their positions. There is very extensive prior work in this area, which makes the novelty somewhat limited. The proposed architecture is evaluated on two types of toy task (including some variants of those tasks): SVRT with/without OOD setting and ART/CLEVR-ART. All tasks use simple synthetic images with highly local structure in entirely uncluttered scenes, and their solutions seems to strongly rely on positions of objects. One would expect any architecture that makes those positions explicit to do exceptionally well on these specific tasks. Training sets for these tasks are also extremely small and will thereby favor architectures that are based on hardcoded (not trained) feature extraction. Given the extensive prior work on foveated vision, how do existing methods perform on these tasks?

**Questions:**

I'm not sure about the term "active", since the architecture relies on a fixed (and not adaptive or recurrent) scheme to extract local features?

Why is the dataset size for results in Table 1 restricted to 500/1000 training examples? This seems arbitrary. Figure 4 seems to suggest that the training set is larger?

Can the SVRT #1-OOD experiment be extended to other subsets than SVRT #1?

Is there a reason why the results in Table 3 (on a subset of ART) do not include all models considered previously (e.g. Attn-ResNet?)

It would be very beneficial to include a modern vision-language model in the evaluation, as these models have become very good at solving similar reasoning tasks.

Some additional related work:

"Learning to combine foveal glimpses with a third-order Boltzmann machine", Larochelle et al. 2010

"Multiple Object Recognition with Visual Attention", Ba et al. 2014

"Show, Attend and Tell: Neural Image Caption Generation with Visual Attention", Xu et al. 2015

---

> ### Author Response · Authors · 2024-11-22
>
> We thank the Reviewer for constructive comments. Below we have prepared detailed responses to the points raised in the Weaknesses and Questions sections.
>
> > **Weakness**: The proposed architecture can be viewed as a local feature extractor that encodes local image regions and their positions. There is very extensive prior work in this area, which makes the novelty somewhat limited.
>
> The motivation for our approach was to take inspiration from the human active vision that does not process the visual input entirely but rather salient/task-relevant parts of it. There is indeed extensive prior work in this area, and we provide a more extensive discussion on that in our response to the last Reviewer’s question. Most of the prior work focused on using reinforcement learning (RL) to train a glimpsing policy. Our approach, in contrast, uses the concept of saliency maps to determine where to look. Lastly, to the best of our knowledge, for the first time such bio-inspired vision approach is extensively evaluated in the context of relational reasoning, sample efficiency and generalization robustness, surpassing baselines operating using more conventional principles.

---

> ### Author Response · Authors · 2024-11-22
>
> > **Weakness**: One would expect any architecture that makes those positions explicit to do exceptionally well on these specific tasks
>
> The spatial information, as the Reviewer points out, is indeed very important. But as shown in our experiments with different downstream architectures (e.g. Table 1-4), it is not sufficient to simply make this information explicit. What our approach provides on top is a way to effectively combine the locally extracted features with their locations for reasoning about relations between different parts of the visual input.

---

> ### Author Response · Authors · 2024-11-22
>
> > **Weakness**: Training sets for these tasks are also extremely small and will thereby favor architectures that are based on hardcoded (not trained) feature extraction
>
> This may not be always true, especially if OOD generalization is evaluated. For example, the state-of-the-art baseline of Slot-Abstractor relies on hardcoded feature extraction stage that was pre-trained on larger datasets. As we show in our results, this model struggles in many OOD settings, see e.g. Table 2 and Figure 7.

---

> ### Author Response · Authors · 2024-11-22
>
> > **Weakness**: Given the extensive prior work on foveated vision, how do existing methods perform on these tasks?
>
> &
>
> > **Question**: Some additional related work:
> > "Learning to combine foveal glimpses with a third-order Boltzmann machine", Larochelle et al. 2010
> > "Multiple Object Recognition with Visual Attention", Ba et al. 2014
> > "Show, Attend and Tell: Neural Image Caption Generation with Visual Attention", Xu et al. 2015
>
> We thank the Reviewer for hints on the relevant prior work. We included the discussion of this work in the revised “Related work” section of our manuscript as follows:
>
> _The approach from (Larochelle & Hinton, 2010) and its ANN-based extension (Mnih et al., 2014)
> use reinforcement learning (RL) to train a glimpsing policy to determine the next glimpse location
> either in a continuous 2D space or in a discrete grid-based space of all possible glimpse locations.
> While this approach was evaluated mainly on simple image classification and object detection tasks,
> Ba et al. (2014); Xu et al. (2015) extended it to more complex images and image captioning tasks.
> However, having been evaluated on several visual reasoning tasks by Vaishnav & Serre (2023), the
> RL-based approaches could not achieve reasonable performance. This is likely caused by learning
> inefficiency in exploring the large space of all possible glimpse locations._

---

> ### Author Response · Authors · 2024-11-22
>
> > **Question**: I'm not sure about the term "active", since the architecture relies on a fixed (and not adaptive or recurrent) scheme to extract local features?
>
> By the term “active” we mean that our model sequentially picks the most important image regions, thereby mimicking the human vision with active eye movements. Our scheme to select those regions can be viewed as adaptive to some extent. Firstly, it does not attend to some pre-defined set of image locations. Instead, those locations depend on the saliency extraction process. Secondly, the selection of the next glimpse location depends on the previously selected locations that are suppressed by the inhibition-of-return (IoR) mechanism. Still, further extensions are possible, such as modulating the choice of locations via top-down connections from deeper layers of the network. This improvement would also allow to incorporate various additional information making the glimpsing behavior potentially more goal-driven.

---

> ### Author Response · Authors · 2024-11-22
>
> > **Question**: Why is the dataset size for results in Table 1 restricted to 500/1000 training examples? This seems arbitrary. Figure 4 seems to suggest that the training set is larger?
>
> The results reported in Table 1 and Figure 4 evaluate the sample efficiency, where the SVRT dataset is deliberately restricted – the smaller, the more difficult the task. The restriction to 500 and 1000 training examples in Table 1 was made to be comparable and consistent with the prior work (Mondal et al., 2024) that demonstrated state-of-the-art results.
> Still, we also felt that this choice appears arbitrary and, therefore, we provided more extensive results in subsequent Figure 4, including both larger and smaller dataset sizes than reported in the prior art.

---

> > ### Comment · Reviewer_8myJ · 2024-11-26
> >
> > The updated Figure 4 suggests that the benefit of the proposed method is clearly just sample efficiency, since all methods seem to converge to perfect accuracy. Further, they do so with a reasonably small number of training examples of probably well below 100k examples.
> >
> > The new results suggest that the conclusions from Table 1 can actually be quite misleading. The table suggests that there is a clear accuracy order with clearly winning and losing methods, but in reality, all results are specific to the arbitrary number of training examples, and all methods converge to perfect accuracy very quickly anyway. I understand that the reasoning for picking 1000 and 500 is prior work, but it still seems entirely arbitrary otherwise. Am I missing something here? Thanks for clarification.

---

> ### Author Response · Authors · 2024-11-22
>
> > **Question**: Can the SVRT #1-OOD experiment be extended to other subsets than SVRT #1?
>
> This would be an interesting experiment. However, the SVRT #1-OOD dataset introduced by (Puebla and Bowers, 2022) is currently available only for SVRT #1. While the code to generate SVRT #1-OOD is publicly available, it is not compatible with the code for the original SVRT. It is therefore not straightforward to extend the SVRT #1-OOD experiment to other SVRT tasks. However, we performed similar experiments using other more complex datasets such as ART and CLEVR-ART that also evaluate OOD generalization.

---

> ### Author Response · Authors · 2024-11-22
>
> > **Question**: Is there a reason why the results in Table 3 (on a subset of ART) do not include all models considered previously (e.g. Attn-ResNet?)
>
> For each table, we selected the best performing models reported in prior work. No prior work reported the performance for Attn-ResNet.

---

> ### Author Response · Authors · 2024-11-22
>
> > **Question**: It would be very beneficial to include a modern vision-language model in the evaluation, as these models have become very good at solving similar reasoning tasks.
>
> Thank you for this suggestion. To address this aspect, we conducted an experiment described in the general comment to all Reviewers “**Study on LLMs’ capabilities of solving same-different task**”

---

> > ### Comment · Area_Chair_vSkP · 2024-11-25
> >
> > Dear Reviewer,
> >
> > The authors have provided their responses. Could you please review them and share your feedback?
> >
> > Thank you!

---

> ### Author Response · Authors · 2024-11-27
>
> [UPDATED]
>
> Firstly, we would like to clarify that Figure 4 has been present in its current form since our initial submission. There was no update or new results related to Table 1 and Figure 4 included during the rebuttal period.
>
> The purpose of Table 1 and Figure 4 is to show the results from evaluating sample efficiency. Although we agree that the sizes of 1000 and 500 samples appear to be arbitrary, they follow the common convention for the SVRT task and immediately enable comparisons with prior papers. To clarify how the table and the figure should be interpreted, we made the following revisions to the current draft:
> - We modified the caption of Table 1: _Test accuracy for SVRT evaluated for common dataset sizes of 1000 and 500 samples_
> - We modified the caption of Figure 4: _Sample efficiency evaluation for SVRT beyond the common dataset sizes of 1000 and 500 samples_
>
> As for conclusions stemming from Table 1 and Figure 4, we would like to point out that:
> - Sample efficiency, shown in Table 1 and Figure 4, is an important benefit of our method, which we emphasize in our manuscript.
> - In most cases, the accuracy order of the models in Table 1 is retained for other dataset sizes shown in Figure 4. Of course, Table 1 should not be considered in isolation, thus we immediately placed Figure 4 after it.
>
> We hope that our motivations are reasonable and provide an acceptable trade-off between maintaining comparability with the reporting in prior papers and still providing accurate insights. Shall this be insufficient, we are happy to incorporate further suggestions.

---

> > ### Comment · Area_Chair_vSkP · 2024-11-30
> >
> > Dear Reviewer,
> >
> > The authors have provided their responses. Could you please review them and share your feedback?
> >
> > Thank you!

---

> > ### Comment · Reviewer_8myJ · 2024-12-02
> >
> > Thank you for the clarifications. I will raise my rating to 6.

---

### Official Review · Reviewer_1Rv1 · 2024-11-03

**Soundness:** 3
**Presentation:** 3
**Contribution:** 3
**Rating:** 6
**Confidence:** 4

**Summary:**

The authors develop a system equipped with a novel Glimpse-based Active Perception (GAP) that sequentially glimpses at the most salient regions of the input image and processes them at high resolution. Their approach reaches state-of-the-art performance on several visual reasoning tasks being more sample-efficient, and generalizing better to out-of-distribution visual inputs than prior models.

**Strengths:**

1. The paper introduces a novel "Glimpse-Based Active Perception" (GAP) model inspired by human active vision.
2. The proposed approach reaches state-of-the-art performance on several visual reasoning tasks being more sample-efficient, and generalizing better to out-of-distribution visual inputs than prior models.

**Weaknesses:**

1.  The images in the four selected datasets seem relatively simple, with very clean backgrounds. Have you considered comparing your proposed model with baseline models on more realistic image datasets, such as the COCO dataset?
2. Given that the GAP mechanism involves multiple steps (e.g., saliency map generation, inhibition of return), have you compared its computational performance with other baseline models?
3. Why can GAP-Abstractor improve OOD generalization of same-different relation and more abstract relations?

**Questions:**

1. Have you explored alternative methods for computing the saliency map, given that GAP's effectiveness largely depends on the map's accuracy in identifying key image regions?
2. What is the impact of using a hard versus a soft mask M(x_t)?

---

> ### Author Response · Authors · 2024-11-22
>
> We thank the Reviewer for the constructive comments. Below we have prepared detailed responses to the points raised in the Weaknesses and Questions sections.
>
> > **Weakness**: The images in the four selected datasets seem relatively simple, with very clean backgrounds. Have you considered comparing your proposed model with baseline models on more realistic image datasets, such as the COCO dataset?
>
> While images in the four selected datasets are simple, it has been shown by prior works, such as (Vaishnav et al., 2023, Mondal et al., 2024), that visual reasoning tasks with such images pose a significant challenge to AI models. Nevertheless, to get insights on the potential of our approach to deal with more complex images, we have conducted an experiment using more realistic objects. The description and results can be found in the general comment to all Reviewers: “**Experiment with more realistic objects and alternative saliency maps**”.

---

> ### Author Response · Authors · 2024-11-22
>
> > **Weakness**: Given that the GAP mechanism involves multiple steps (e.g., saliency map generation, inhibition of return), have you compared its computational performance with other baseline models?
>
> &
>
> > **Question**: Have you explored alternative methods for computing the saliency map, given that GAP's effectiveness largely depends on the map's accuracy in identifying key image regions?
>
> To address this important aspect, we conducted additional experiments exploring different ways to compute the saliency maps. The description and results can be found in the general comment to all Reviewers: “**Experiment with more realistic objects and alternative saliency maps**”.

---

> ### Author Response · Authors · 2024-11-22
>
> > **Weakness**: Why can GAP-Abstractor improve OOD generalization of same-different relation and more abstract relations?
>
> To provide more insights on this aspect, we revised our “Discussion” section (Sec. 6):
>
> _Our results suggest that factorizing an image into its complementary “what” and “where” contents
> plays an essential role. The low-dimensional “where” content (glimpse locations) is crucial since it
> does not contain any visual information and, in turn, allows to learn abstract relations that are agnostic to specific visual details. To capitalize on that, it is important to process the “where” content explicitly, disentangling it from its “what” complement. We implemented this by using the recently introduced relational cross-attention mechanism (employed by the Abstractor downstream architecture) where the “what” content is used to compute the attention scores for the processing of the “where” content. In contrast, implicitly mixing the “what” and the “where” contents, as in the case of the Transformer downstream architecture, weakens the generalization capabilities. This can be seen by comparing the performance between GAP-Transformer and GAP-Abstractor. To further support this, we provide supplementary results in Appendix D.2 showing the importance of using both the “what” and the “where” contents instead of just one of them for task solving._
>
> _Another important aspect of our model is that it processes only the most salient image regions while
> ignoring the unimportant ones. The inferior performance of models where the downstream archi-
> tectures receive full information (i.e. all patches that constitute the image) suggests that supplying
> a superfluous amount of information may distract the models hindering the learning process. Additional results provided in Appendix D.3 elucidate this further. Specifically, we show that it is
> insufficient to naively reduce the amount of information provided to the downstream architecture,
> by discarding uninformative image parts. Instead, it has to be ensured that the supplied information
> is well structured. In the context of GAP, it means that the glimpse contents have to be centered
> around salient image regions. For example, the image patches of the glimpse contents should contain objects’ edges in their central parts rather than somewhere in the periphery._

---

> ### Author Response · Authors · 2024-11-22
>
> > **Question**: What is the impact of using a hard versus a soft mask M(x_t)?
>
> A soft mask was motivated by a hypothesis that in complex scenes (such as CLEVR-ART) it can be beneficial to make several glimpses close to each other. The hard mask does not allow for that, as it masks out the area around the glimpse location in the saliency map, thereby excluding any further glimpsing in this area.
> We tested the soft mask only on CLEVR-ART tasks and it resulted in accuracy difference within 1-2% for our GAP-Abstractor model, see the table below
>
> | Mask type |     RMTS      |      ID      |
> | :-------: | :-----------: | :----------: |
> |   hard    | 93\.6 ± 0.9 | 93\.4 ± 1.3  |
> |   soft    | 95\.9 ± 1.4 | 92\.1 ± 1.1  |

---

> > ### Comment · Area_Chair_vSkP · 2024-11-25
> >
> > Dear Reviewer,
> >
> > The authors have provided their responses. Could you please review them and share your feedback?
> >
> > Thank you!

---

> ### Comment · Reviewer_1Rv1 · 2024-11-25
>
> Thank you for the author's response. I would like to maintain my score. My primary concern is that the images in the datasets used, including the newly added Super-CLEVR dataset, lack sufficient realism. This makes me skeptical about the practical applicability of the proposed approach in real-world scenarios.

---

### Official Review · Reviewer_xwHi · 2024-11-04

**Soundness:** 3
**Presentation:** 3
**Contribution:** 3
**Rating:** 8
**Confidence:** 3

**Summary:**

This paper proposes a novel method for visual reasoning, called Glimpse-based Active Perception (GAP). Based on that the human eye selectively concentrates on salient and/or task-relevant parts of a scene, the authors devise a method that extracts salient regions of an image and feeds them into downstream architectures to enforce perception of salient regions only. Firstly, a saliency map is built based on error neurons, which marks salient regions as those that differ significantly with their surrounding neighbours. Secondly, salient regions are extracted from the image, using the saliency map, with either the multi-scale or log-polar glimpse sensor. Lastly, these salient regions and their locations are fed into either Transformer or Abstractor to perform visual reasoning tasks, each dubbed as GAP-Transformer and GAP-Abstractor. By forcing the downstream architectures to concentrate on salient regions only, the GAP-Abstractor achieves SOTA or comparative-to-previous-SOTA performance on the conventional visual reasoning benchmarks, OOD benchmarks, and real-image-based OOD benchmarks.

**Strengths:**

- Authors propose a well-motivated method:
  - They explain the motivation behind GAP logically, linking back to the human perception model throughout their method section.
  - Ie. They explain that they concentrate on salient regions since "humans use active vision"; They explain that they use the concept of error neurons since "the activity of neurons is influenced ... also by stimuli that come from the surroundings of those receptive fields".

- The proposed method attains strong performance on benchmarks:
  - Especially compared to previous work that do not use pretraining as GAP, GAP-Abstractor outperforms them by large margins.
  - Authors also infer on four datasets, each with different purposes (testing visual reasoning only vs OOD generalization too) and different domains (binary figures vs real-world images).

- The paper is well-written and easy to follow with both textual and visual explanations.

- This paper introduces a novel and logical method to field of Visual Reasoning, which is of great importance since visual reasoning is required for a range of real-life tasks, and thereby contributes to the community.

**Weaknesses:**

- Authors do not make it explicit or clear that different sensors (multi-scale or log-polar) have been used across different datasets for the same downstream architectures (ViT/Abstractor). Ie. For SVRT, GAP-Abstractor used the multi-scale sensor but for SVRT #1-OOD it used the log-polar sensor. This should be made clear in the main paper, instead of referring to the Appendix.
  - Also, in Appendix D, authors do not explain why each sensor works better or worse for each downstream architecture-dataset pair. Since the performance depends on the type of sensor and since authors have proposed to use both, authors should explain where each sensor may be a better choice.

- References are missing in Tables and Figures.
- Explain what the tasks RMTS and ID are, at least briefly, in Section 5.4 before explaining the results.

**Questions:**

- Can the authors provide an explanation to why GAP-Abstractor perform worse on 'straight lines' and 'rectangles' compared to other classes?
- Since the authors explained that "glimpsing behavior can be distracted by spurious salient locations such as edges of shades or regions of high reflection", would performing GAP (finding salient regions) on feature maps, instead of images, where these spurious features are probably less influential, increase performance? Can authors provide results?
- Please explain why each sensor works better or worse for each downstream architecture-dataset pair.

---

> ### Author Response · Authors · 2024-11-22
>
> We thank the Reviewer for the constructive comments. Below we have prepared detailed responses to the points raised in the Weaknesses and Questions sections.
>
> > **Question**: Can the authors provide an explanation to why GAP-Abstractor perform worse on 'straight lines' and 'rectangles' compared to other classes?
>
> The reason that GAP-Abstractor performs worse on ‘straight_lines’ and ‘rectangles’ is that the dataset contains many images where the straight lines and rectangles are very similar to each other with only slight differences in size. We included Figure B.3 into the revised Appendix that shows a few examples with different shapes where even humans may have difficulties to tell the difference.

---

> ### Author Response · Authors · 2024-11-22
>
> > **Question**: Since the authors explained that "glimpsing behavior can be distracted by spurious salient locations such as edges of shades or regions of high reflection", would performing GAP (finding salient regions) on feature maps, instead of images, where these spurious features are probably less influential, increase performance? Can authors provide results?
>
> We thank the reviewer for the suggestion of improving the glimpsing behavior by applying GAP on top of the feature maps instead of the images directly. To address this question, we conducted additional experiments described in the general comment “**Experiment with more realistic objects and alternative saliency maps**”

---

> ### Author Response · Authors · 2024-11-22
>
> > **Question**: Please explain why each sensor works better or worse for each downstream architecture-dataset pair.
>
> &
>
> > **Weakness**: Authors do not make it explicit or clear that different sensors (multi-scale or log-polar) have been used across different datasets for the same downstream architectures (ViT/Abstractor). Ie. For SVRT, GAP-Abstractor used the multi-scale sensor but for SVRT #1-OOD it used the log-polar sensor. This should be made clear in the main paper, instead of referring to the Appendix
>
> We clarified the usage of different sensors in the main text at the beginning of the “Results” section (Sec. 5) as follows:
>
> _[…]
> One important hyper-parameter is the type of glimpse sensor (multi-scale or log-polar) that we select for each of the four considered datasets. We  report performance for different glimpse sensors in Appendix D._
>
> In addition, we highlighted the best-performing sensors in tables Table D.1-D.4 and revised the Appendix D to comment on the performance differences for each downstream architecture-dataset pair in Sec. D.1 as follows:
>
> _We evaluated two different glimpse sensors for each of the four datasets considered in this work, see
> Table D.1- D.4 and Figure D.6. For the Abstractor-based downstream architecture, we observe com-
> parable performance for both sensors in most cases. The most significant performance difference is observed
> for SVRT #1-OOD and two tasks from the ART dataset (Table D.2-D.3) where the log-polar sensor
> achieves around 5% better accuracy. For the Transformer-based downstream architecture, the per-
> formance difference between two glimpse sensors averaged over all tasks is around 5%. However,
> there are tasks where the multi-scale sensor results in around 10% worse accuracy, see Table D.1,
> and 10% better accuracy, see Table D.3. Overall, based on the performance data, there is no clear
> quantitative trend favoring either of the two glimpse sensors. We ascribe this to the qualitative ad-
> vantages and disadvantages of both sensors. In particular, the log-polar sensor benefits from the properties of the log-polar space where the rotation and scaling in the cartesian space become trans-
> lations, see e.g. (Javier Traver & Bernardino, 2010). However, the warping into log-polar space may
> make it difficult to capture information about image parts that are far from the glimpse locations.
> The multi-scale sensor, in contrast, captures the distant information more faithfully using downsam-
> pled image patches of different sizes. The disadvantage of this, however, is that the resulting glimpse
> content contains additional dimensions for the different scales._

---

> ### Author Response · Authors · 2024-11-22
>
> > **Weakness**: References are missing in Tables and Figures.
>
> &
>
> > **Weakness**: Explain what the tasks RMTS and ID are, at least briefly, in Section 5.4 before explaining the results.
>
> We thank the Reviewer for the suggestions.
>
> We attempted to introduce references in all tables and figures, but due to long reference formatting of ICLR, the material would not fit the page width and the maximum page count. The references to all baseline models are briefly summarized in Sec. 4 in the "Baselines" paragraph.
>
> We provided a brief description of the RMTS and ID tasks at the beginning of Section 5.4 as follows:
>
> _We test our approach on two tasks from ART – relational-match-to-sample (RMTS) and identity rules (ID) – but with more realistic images using the CLEVR-ART dataset. In the RMTS task the model has to determine whether the objects in the top row are in the same ”same/different” relation as the objects in the bottom row. In the ID task the model has to determine whether bottom row contains objects that follow the same abstract pattern (ABA, ABB or AAA) as the objects in the top row. Figure B.4 provides examples for the two tasks_

---

> > ### Comment · Area_Chair_vSkP · 2024-11-25
> >
> > Dear Reviewer,
> >
> > The authors have provided their responses. Could you please review them and share your feedback?
> >
> > Thank you!

---

> > > ### Comment · Reviewer_xwHi · 2024-11-26
> > >
> > > I suggest authors to include a more in-depth analysis (ie. qualitative visualizations) on where each glimpse sensor works well too.
> > > However, most of my concerns are resolved. I will raise my score.

---

### Official Review · Reviewer_YJPp · 2024-11-07

**Soundness:** 3
**Presentation:** 3
**Contribution:** 3
**Rating:** 8
**Confidence:** 4

**Summary:**

The paper focuses on understanding visual relations. This remains a challenging problem for current vision models. To deal with this challenge the paper leverages active vision where the learning of visual relations is grounded in actions that we take to fixate objects and their parts by moving our eyes. The proposed approach with glimpse-based active perception demonstrates promising performance on a range of visual reasoning tasks.

**Strengths:**

* The paper provides interesting insights into visual reasoning problems.
* The proposed glimpse-based active perception is relatively novel and interesting.
* The proposed approach shows promising performance on a range of visual reasoning problems.
* The paper considers diverse visual "sensors" and downstream architectures.

**Weaknesses:**

* While the proposed approach is somewhat novel it is similar to prior work such as "AdaGlimpse: Active Visual Exploration with
Arbitrary Glimpse Position and Scale, ECCV 2024" which also focus on what and where to look.

* Current state of the art vision-language models such as LLaVA (NeurIPS 2024) keep the visual features from the target image in the context window. This means that they can actively attend to the image as many times are necessary to extract visual features. It would be interesting to compare the proposed approach to current SOTA VLMs such as LLaVA.

* Prior work such as "Look, Remember and Reason: Grounded reasoning in videos with language models, ICLR 2024" used surrogate tasks to decide what and where to look at. It would be beneficial to include a discussion of this related work.

* Synthetic data: Three out of the four datasets used for evaluation are based on synthetic datasets. It would be beneficial to include more real world datasets such as GQA or Super-Clevr.

**Questions:**

* The paper should include a broader discussion of related work (see above).
* The paper should better motivate the choice of evaluation datasets.

---

> ### Author Response · Authors · 2024-11-22
>
> We thank the Reviewer for the constructive comments. Below we have prepared detailed responses to the points raised in the Weaknesses and Questions sections.
>
> > **Weakness**: While the proposed approach is somewhat novel it is similar to prior work such as "AdaGlimpse: Active Visual Exploration with Arbitrary Glimpse Position and Scale, ECCV 2024" which also focus on what and where to look.
>
> &
>
> > **Weakness**: Prior work such as "Look, Remember and Reason: Grounded reasoning in videos with language models, ICLR 2024" used surrogate tasks to decide what and where to look at. It would be beneficial to include a discussion of this related work.
>
> &
>
> > **Question**: The paper should include a broader discussion of related work (see above).
>
> We thank the Reviewer for the suggestions on related papers. We included them into Section 3, “Related Work”, with the following revisions:
>
> _The approach from (Larochelle & Hinton, 2010) and its ANN-based extension (Mnih et al., 2014)
> use reinforcement learning (RL) to train a glimpsing policy to determine the next glimpse location
> either in a continuous 2D space or in a discrete grid-based space of all possible glimpse locations.
> While this approach was evaluated mainly on simple image classification and object detection tasks,
> Ba et al. (2014); Xu et al. (2015) extended it to more complex images and image captioning tasks.
> However, having been evaluated on several visual reasoning tasks by Vaishnav & Serre (2023), the
> RL-based approaches could not achieve reasonable performance. This is likely caused by learning
> inefficiency in exploring the large space of all possible glimpse locations. Nevertheless, the RL-based approaches that use more efficient RL techniques such as (Pardyl et al., 2025) to learn complex glimpsing policies are relevant to our work as they can be integrated into our model to enhance its capabilities of dealing with real-world images. Our approach, in contrast, leverages the concept of saliency maps to determine the next glimpse location which significantly reduces the space of glimpse locations to the most salient ones.
> […]
> Interestingly, in the domain of visually extended large language models (LLMs), Bhattacharyya et al. (2024) showed that forcing these models via surrogate tasks to collect relevant pieces of information before solving the actual task improves the final performance. While using a different, compared to ours, approach of surrogate tasks to achieve that, this work points to a potential of integrating our conceptual approach into LLMs._

---

> ### Author Response · Authors · 2024-11-22
>
> > **Weakness**: Current state of the art vision-language models such as LLaVA (NeurIPS 2024) keep the visual features from the target image in the context window. This means that they can actively attend to the image as many times are necessary to extract visual features. It would be interesting to compare the proposed approach to current SOTA VLMs such as LLaVA.
>
> We thank the Reviewer for the suggestion on the VLM comparison. We evaluated one of the current SOTA VLMs, Pixtral (Agrawal et al., 2024), on the same-different task and provided the results in the general comment to all reviewers: “Study on LLMs’ capabilities of solving same-different task”
>
> References
> - Agrawal, Pravesh, Szymon Antoniak, Emma Bou Hanna, Baptiste Bout, Devendra Chaplot, Jessica Chudnovsky, Diogo Costa et al. "Pixtral 12B." arXiv preprint arXiv:2410.07073 (2024).

---

> ### Author Response · Authors · 2024-11-22
>
> > **Weakness**: Synthetic data: Three out of the four datasets used for evaluation are based on synthetic datasets. It would be beneficial to include more real world datasets such as GQA or Super-Clevr.
>
> &
>
> > **Question**:  The paper should better motivate the choice of evaluation datasets.
>
> It has been shown by prior works, such as (Vaishnav et al., 2023, Mondal et al., 2024), that visual reasoning tasks with simplistic images can pose significant challenges to AI models. Moreover, as we show in the general comment “Study on LLMs’ capabilities of solving same-different task”, even VLMs appear to encounter difficulties. Therefore, solving these visual reasoning tasks presents an important research step forward. We clarified this motivation in the revised manuscript in Sec. 4 in the paragraph on Datasets with the following revision:
>
> _[…] Finally, we test the model’s potential to scale to more complex images by testing it on the CLEVR-ART dataset (Webb et al., 2024a), see Figure 3(d). It contains two tasks from ART but with more realistic images. Each dataset is described in more detail in Appendix B.
> While all tasks described above consist of simple images, they are still challenging, as illustrated by the limited accuracy of the state-of-the-art models described below._
>
> Despite the importance of the currently selected datasets, we agree with the Reviewer that it is important to explore more realistic images, and we thank the Reviewer for the suggestion on the datasets.  Although at this point our model does not include an NLP module, we have conducted an experiment using the more realistic objects from the suggested Super-CLEVR dataset. The description and results can be found in the general comment to all Reviewers: “**Experiment with more realistic objects and alternative saliency maps**”.

---

> > ### Comment · Area_Chair_vSkP · 2024-11-25
> >
> > Dear Reviewer,
> >
> > The authors have provided their responses. Could you please review them and share your feedback?
> >
> > Thank you!

---

> > ### Comment · Reviewer_YJPp · 2024-11-25
> > **Good rebuttal**
> >
> > The rebuttal address most of my concerns. I will raise my score.

---

### Author Response · Authors · 2024-11-22
**Study on LLMs’ capabilities of solving same-different task**

To address the common questions of reviewers about how well visual LLMs (VLMs) solve the visual reasoning tasks considered in our work, we conducted an experiment with Pixtral 12B model (Agrawal et al., 2024). We chose this recently introduced open-source model due to its superior performance in various visual reasoning benchmarks.

We evaluated Pixtral on the same-different task from the SVRT dataset where the model has to determine whether two shapes are the same up to translation. Below we list the accuracy for different prompts and report the best performance in a table along with the performance of our GAP-Abstractor model on these images for comparison (this value corresponds to the first bar on the left in Figure D.6).

- prompt 1: 62%
	- Answer the following question using only information from the image with "yes" or "no". Question: Are the two shapes the same up to translation?
- prompt 2: 63%
	- Are the two shapes in the image the same up to translation? Answer with "yes" or "no".
- prompt 3: 67%
	- Are the two shapes in the image the same? Please answer "yes" or "no".
- prompt 4: 71%
	- Is the second shape in the image an identical copy of the first shape just drawn at a different location? Please answer "yes" or "no".
- prompt 5: 67%
	- You see an image from SVRT dataset that tests visual reasoning capabilities. Answer the following question using only information from the image with "yes" or "no". Question: Are the two shapes the same up to translation?
- prompt 6: 58%
	- Given an image with two shapes, you have to say 'yes' if they are the same up to translation or 'no' if they are different. The first example image contains two shapes that are the same and the second example contains shape that are different. Given now the following image with two shapes, are they the same up to translation? Please answer 'yes' or 'no'
		- Note: This prompt included two images as examples for the “same” and “different” cases

|              Model               | Accuracy |
| :------------------------------: | :------: |
|          GAP-Abstractor          |   99%    |
| Pixtral (best performing prompt) |   71%    |

Importantly, a related work mentioned by one of the reviewers (Bhattacharyya et al., 2024) shows that “guiding” a VLM to sequentially attend to the relevant parts of the visual input can improve its reasoning capabilities. Hence, our approach has the potential to be combined with VLMs to provide such guidance through the glimpsing.

References
- Agrawal, Pravesh, Szymon Antoniak, Emma Bou Hanna, Devendra Chaplot, Jessica Chudnovsky, Saurabh Garg, Theophile Gervet et al. "Pixtral 12B." arXiv preprint arXiv:2410.07073 (2024).
- Bhattacharyya, Apratim, et al. "Look, Remember and Reason: Grounded Reasoning in Videos with Language Models." _The Twelfth International Conference on Learning Representations, 2024_.

---

### Author Response · Authors · 2024-11-22
**Experiment with more realistic objects and alternative saliency maps**

We evaluated our model on more realistic objects. Additionally, we show how different saliency maps may improve the model’s performance in this case.

In particular, we used the objects from the Super-CLEVR dataset (Li et al., 2023), including cars, planes and motorbikes, to generate a new test set for the Identity Rules (ID) task from the CLEVR-ART dataset. The task is to determine whether three objects in the top row are arranged under the same abstract rule as the objects the bottom row. For example, objects “bus”, “car” and “bus” in one row are arranged under the abstract rule ABA just as objects “car”, “bus” and “car.  We considered four models trained on the original CLEVR-ART train set that consists only of simple cubes (Figure 6), and evaluated them on the new dataset, referred to as Super-CLEVR test set. The example images from the Super-CLEVR test set are shown in Figure E.8 in the revised manuscript.

Simultaneously, we explored using different saliency maps. The first model is our GAP-Abstractor model with the original error neurons applied on the raw image to extract the saliency map, we refer to this model here as GAP-Abstractor (EN). The second model, referred to as GAP-Abstractor (CNN+EN), is a modification of GAP-Abstractor that extracts the saliency map by applying the error neurons on the feature maps computed by the first layer of ResNet-18 that was pre-trained on ImageNet. As suggested by one of the Reviewers, computing the saliency map from the feature maps should make the glimpsing behavior less distracted by spurious salient locations such as edges of shades or regions of high reflection. The third model, referred to as GAP-Abstractor (IK), employs an alternative way to extract the saliency maps using a more complex bio-plausible model proposed by (Itti and Koch, 1998). The example saliency maps are shown in Figure E.8.

The table below shows the corresponding results for five runs with the best and second-best accuracy being in bold and italics, respectively.  Slot-Abstractor is the best performing prior art providing the baseline.
| Model                       |  original  test set | Super-CLEVR  test set  |
| :-------------------------- | :----------------------: | :-------------------------: |
| Slot-Abstractor (prior art) | 91\.61 ± 0.2             | 24\.1 ± 0.8                 |
| GAP-Abstractor (EN)         | **93\.4 ± 1.3**          | 72\.3 ± 2.3                 |
| GAP-Abstractor (CNN+EN)     | _92\.7 ± 1.8_              | _80\.2 ± 1.9_                 |
| GAP-Abstractor (IK)         | 90\.2 ± 2.1              | **82\.1 ± 2.2**             |

As it can be seen, Slot+Abstractor fails completely on the unseen Super-CLEVR objects while the performance of our GAP-Abstractor does not degrade so severely. Moreover, we see that GAP-Abstractor (CNN+EN) performs better on Super-CLEVR test set while performing comparably on the original test set. Therefore, applying the error neurons on top of features maps rather than on top of the raw images results in a saliency map that can better drive the glimpsing behavior.
Lastly, we see that GAP-Abstractor (IK) with an alternative bio-plausible way to extract the saliency map reaches the best performance on the Super-CLEVR test set while being slightly worse on the original test set. A possible reason for this difference is that this saliency map extractor was designed to handle real-world scenes which are of much higher visual complexity compared to simple shapes. Overall, these results indicate a possible direction of enhancing our approach for visual reasoning by integrating more powerful saliency maps.

References
- Li, Zhuowan, Xingrui Wang, Elias Stengel-Eskin, Adam Kortylewski, Wufei Ma, Benjamin Van Durme, and Alan L. Yuille. "Super-clevr: A virtual benchmark to diagnose domain robustness in visual reasoning." In Proceedings of the IEEE/CVF Conference on Computer Vision and Pattern Recognition, pp. 14963-14973. 2023.
- Itti, L., Koch, C., & Niebur, E. (1998). A model of saliency-based visual attention for rapid scene analysis. IEEE Transactions on Pattern Analysis and Machine Intelligence, 20(11), 1254–1259. IEEE Transactions on Pattern Analysis and Machine Intelligence.

---

### Author Response · Authors · 2024-12-03

We sincerely thank the reviewers for their time and effort in thoroughly evaluating our submission. We greatly appreciate their thoughtful feedback and constructive suggestions, which have significantly contributed to improving the quality of our paper.

---

### Meta-Review · Area_Chair_vSkP · 2024-12-13

**Metareview:**

The paper proposes a Glimpse-based Active Perception (GAP) system that sequentially focuses on salient image regions, leveraging spatial information from eye-movement-inspired actions to represent visual relations, achieving state-of-the-art performance in visual reasoning tasks with improved sample efficiency and out-of-distribution generalization.

The manuscript is well-written, presenting an innovative method with convincing results that surpass the current state-of-the-art, while offering valuable insights into plausible mechanisms for visual reasoning on object relations.

Key concerns, including the use of more realistic image datasets and alternative saliency map computations, were satisfactorily addressed during the rebuttal.

Following the rebuttal, all reviewers agreed to accept the paper.

**Additional Comments On Reviewer Discussion:**

The authors successfully addressed the reviewers' key concerns regarding the use of more realistic image datasets and alternative saliency map computations. The additional experiments provided during the rebuttal effectively clarified issues related to evaluating VLMs on various visual reasoning tasks.

Moreover, the authors addressed specific reviewer questions, including clarifications on the RMTS and ID tasks, missing references in tables, discussions on the intuition behind how GAP-Abstractor enhances OOD generalization for abstract relations, and omissions of several related works such as AdaGlimpse and Look, Remember, and Reason from ICLR2024, among others.

These comprehensive responses satisfactorily resolved the reviewers' initial concerns.

---

### Decision · Program_Chairs · 2025-01-22

Accept (Poster)